# Cross-ancestral GWAS identifies 29 variants across head and neck cancer subsites

Head and neck squamous cell carcinoma (HNSCC) includes diverse cancers arising in the oral cavity, oropharynx, and larynx, with the main risk factors being environmental exposures such as tobacco, alcohol, and human papillomavirus (HPV) infection. The genetic factors contributing to susceptibility across different populations and tumour subsites remain incompletely understood. Here we show, through a genome-wide association and fine mapping study of over 19,000 HNSCC cases and 38,000 controls from multiple ancestries, 18 genetic risk variants and 11 signals from fine mapping of the human leukocyte antigen (HLA) region, all previously unreported. rs78378222, a regulatory variant for *TP53* is associated with a 40% reduction in overall HNSCC risk. We also identify gene-environment interactions, with *BRCA2* and *ADH1B* variants showing effects modified by smoking and alcohol use. Subsite-specific analysis of the HLA region reveals distinct immune-related associations across HPV-positive and HPV-negative tumours. These findings refine the genetic architecture of HNSCC and highlight mechanisms linking inherited variation, immunity, and environmental exposures.

Head and neck squamous cell carcinomas (HNSCC) are a heterogeneous group of cancers originating primarily in the oral cavity (OC), oropharynx (OPC), larynx (LA) and hypopharynx (HPC). Currently, HNSCC is ranked the 6th most common cancer globally, although incidence is predicted to increase 30% by 2030[1,2]. Tobacco smoking and alcohol consumption are major risk factors, particularly in high-income countries, contributing to 72% of cases when used together, while betel quid/areca nut products significantly increase risk in some Asia-Pacific populations[2]. HNSCC subsites can be differentially affected, with smoking more strongly linked to laryngeal cancer and drinking more strongly linked to OC/OPC[3]. There has been a decline in smoking in high-income countries; as such, the increasing incidence could be due to changes in aetiology[4]. Infection with human papillomavirus (HPV), particularly HPV type 16, is a recently identified causal risk factor for OPC[5–7] and the proportion of HPV-associated OPCs is highest in high-income countries (63%–85%)[8]. Disparities in epidemiology, risk, and prognosis highlight the recognition of HPV-associated OPC as a distinct biological entity[9].

Although a limited number of genome-wide association studies (GWAS) have been conducted on HNSCC, a germline contribution to HNSCC risk has been established, with multiple susceptibility loci associated with risk. These include the 4q23 locus (*ADH1B, ADH7*) linked to genes involved in alcohol metabolism, the 5p15.33 locus (*TERT-CLPTM1L*) associated with genes responsible for DNA stability maintenance, and the 6p21 and 6p22 loci, mostly within the human leucocyte antigen (HLA) region, corresponding with genes regulating the innate immune response[10–14]. The 6p locus within the HLA region has been a specific area of focus for HPV-driven cancers, with the hypothesis being that variants influencing immune response to viral antigens would be most relevant for risk[10,12,13]. However, there is an emerging role for the immune microenvironment for other HNSCC subsites[15], suggesting that the HLA may confer risk separately in other HNSCC subsites, potentially via non-HPV mechanisms. Previous GWAS were limited in sample size for HNSCC subsites, making inference between subsites, particularly for HLA, difficult. They were also conducted predominantly in subjects of European ancestry, limiting generalisability of findings[10–14].

Despite knowledge of the major risk factors and several risk loci for HNSCC, identifying those who will develop cancer is still difficult. Not all smokers develop cancer, and risk loci only offer a fractional

✉ e-mail: viranis@iarc.who.int; tom.dudding@bristol.ac.uk

change in risk at the population level. The interaction between environmental factors and risk loci may help explain additional risk and has been reported for lung cancer (smoking)[16], colorectal cancer (alcohol)[17] and bladder cancer (arsenic exposure)[18], among others. Studies investigating these interactions need large sample sizes and individual-level exposure data harmonised across studies, which is often not possible in large GWAS meta-analyses.

Here, we perform a cross-ancestry GWAS of HNSCC using individual-level data, bringing together studies from Europe, North America, South America, South Asia and the Middle East. We identify multiple genetic risk susceptibility loci, determine shared and unique risk loci across subsites, explore interactions between genetic and environmental factors in HNSCC risk and conduct fine mapping of the HLA region. This work lays the foundation for identifying HNSCC susceptibility loci with increased representation from non-European populations.

## Results

### Cross-ancestral meta-analysis identifies 18 novel genetic loci across HNSCC subsites

In this cross-ancestral meta-analysis of two pooled individual-level datasets (Supplementary Data 1), we evaluated 13,092,551 genetic variants in 19,073 HNSCC cases and 38,857 controls. Of the HNSCC cases, there were 5596 (29%) oral cavity (OC), 5411 (28%) oropharyngeal (OPC), 4409 (23%) laryngeal (LA), 898 (5%) hypopharyngeal (HPC), 2759 (14%) unknown (either unknown primary site or not available) or overlapping sites. HPV status was available for 68% of OPC cases, of which 3685 (60%) were HPV(+) (Supplementary Data 2).

We identified and annotated 18 genome-wide associated variants, including two specific to non-European ancestry (Table 1, Fig. 1 and Figure S1) and validated 6 previously identified loci (Supplementary Data 3–5). Fine mapping of the HLA region identified 11 further variants. The SNP based heritability for HNSCC overall was 6.9% (95% Confidence Interval (CI): 4.3, 9.4). Across subsites, heritability ranged from 2.3% (95% CI: 0.0, 4.8) for HPC to 6.3% (95% CI: 1.6, 11.0) for HPV(+) OPC (Supplementary Data 6).

For overall HNSCC, two cross-ancestral variants in the 1q32 region were identified. rs61817953, near *PIK3C2B*, was associated with decreased risk (OR (95%CI) = 0.90 (0.87, 0.93), $p_{meta} = 2.17 \times 10^{-8}$) and rs6679311 near *MDM4*, a strong negative regulator of p53, was associated with increased risk (OR (95% CI) = 1.11 (1.07, 1.14), $p_{meta} = 1.25 \times 10^{-10}$) (Figure S2). The latter is in moderately high LD ($r^2 = 0.75$) with rs4245739, an *MDM4* 3' UTR variant known to increase breast[19] and prostate[20] cancer risk. At the 13q13 locus, rs7334543, a cross ancestral 3' UTR variant in *BRCA2* was associated with decreased risk of overall HNSCC (OR (95%CI) = 0.91 (0.88, 0.94), $p_{meta} = 2.39 \times 10^{-8}$) and was independent from rs11571833, a stop gain variant previously identified in this region for UADTs[14]. Within those of European ancestry, rs78378222 a 3' UTR variant in *TP53*, was associated with a reduced risk of HNSCC overall, (OR (95% CI) = 0.62 (0.52, 0.73), $p = 2.16 \times 10^{-8}$) (Fig. 2a). The effect seemed to be mostly driven by OC and LA (Fig. 2b). The T > G allele frequency of rs78378222 is 0.01 in European (EUR), 0.002 in African (AFR) and American (AMR) populations, and nearly absent in all other 1000 Genomes super-populations; as such, there was no effect of this variant within the Mixed ancestry group. Given its low frequency, technical validation was performed in 2370 samples, and concordance with imputed data was 99.9% (Table S1). There was strong evidence for this variant modulating *TP53* gene regulation, at transcriptional and post-transcriptional levels as indicated by Expression quantitative trait loci (eQTL) and splicing QTLs (sQTL) analyses, with decreased *TP53* expression correlating with a reduced risk of overall HNSCC (Fig. 2c) (Supplementary Data 7). This variant is in the poly-adenylation signal of the *TP53* gene and potentially leads to impaired 3' end processing of *TP53* mRNA[21]. rs78378222 is located within a highly conserved sequence (TTTTATTGTAAAATA ->

TTGTATTGTAAAATA) that appears to be crucial for microRNAs (miRNAs) binding. This region is predicted to interact with 5 different miRNAs (https://dianalab.e-ce.uth.gr/tarbasev9) (Fig. 2d).

For OC, three loci were identified (Table 1). First, rs28419191, an intergenic variant at 5q31 associated with an increased risk of OC (OR (95% CI) = 1.23 (1.15, 1.31), $p_{meta} = 3.16 \times 10^{-10}$) cross ancestrally. This variant was in high LD with rs1131769 ($r^2 = 0.93$), a missense variant in *STING1* which was an identified locus for overall HNSCC risk (OR = 1.13 (1.09, 1.18), $p_{meta} = 2.38 \times 10^{-10}$), and seemed to be driven by non-HPV driven tumours (Table 1 and Fig. 3a, b). Both rs28419191 and rs1131769 correlated with expression of catenin alpha 1 (*CTNNA1*), a gene related to RNA and actin filament binding, but not *STING1* expression in whole blood; as such, the function of this variant is unclear (Fig. 3c). The second cross ancestral variant rs67351073, located at 20q13 in Zinc Finger CCCH-type and G-patch domain containing (*ZGPAT*), was associated with reduced risk of OC (OR (95%CI) = 0.78 (0.72, 0.85), $p_{meta} = 4.45 \times 10^{-8}$). A highly correlated variant, rs4809325 ($r^2 = 0.97$), identified exclusively in the European ancestry, also decreased OC risk. This risk-decreasing variant was correlated with decreased expression of *ZGPAT*, increased *LIME1 and SLC2A4RG* expression and alternative splicing of *LIME1* (Fig. S3 and Supplementary Data 7). Finally, a rare European ancestry-specific intronic variant, rs577454702, located in the mitogen-activated protein kinase 1 (*MAPK1*) gene at 22q11, was associated with a large increased risk of OC (OR (95%CI) = 2.60 (1.86, 3.65), $p = 2.53 \times 10^{-8}$).

For laryngeal cancer, rs55831773, a cross ancestral splice variant, mapping to *ATP1B2* was associated with increased risk (OR (95% CI) = 1.21 (1.13, 1.29), $p_{meta} = 5.1 \times 10^{-9}$). *ATP1B2* is near *TP53*, but conditional analyses suggest this variant is independent of the rare *TP53* 3' UTR variant described for overall HNSCC (Fig. S4b). There was also no evidence that rs55831773 alters *TP53* expression, further suggesting independent effects of these two variants (Fig. S4 and Supplementary Data 7). An intronic variant, rs10419397, located in a gene-dense region of 19p13 was also strongly associated with LA (OR (95%CI) = 1.13 (1.10, 1.17), $p_{meta} = 1.21 \times 10^{-14}$) cross ancestrally. This variant has been found to associate with mitochondrial dysfunction[22,23] and is in very high LD with several variants associated with risk of other cancers, including rs4808616 ($r^2 > 0.99$), a 3' UTR for *ABHD8* linked to breast and lung cancers[24]. rs200410709 is a variant which showed increased risks in the Mixed ancestry but with no evidence of effect in Europeans. It is a deletion variant in an intergenic region, adjacent to the Syntaxin Binding Protein 6 (*STXBP6*) gene (14q12), and was associated with increased risk of LA (OR (95% CI) = 3.38 (2.26, 5.07), $p = 3.57 \times 10^{-9}$) (Fig. S5).

Five HPC specific variants were identified. rs138707495, a rare (MAF: European=0.009, Mixed=0.005) variant located in the 3' UTR of *GDF7* (OR (95%CI) = 3.06 (2.07, 4.53), $p_{meta} = 2.33 \times 10^{-8}$), rs77750788 at 11q25 near *IGSF9B* (OR (95%CI) = 2.07 (1.61, 2.68), $p_{meta} = 2.03 \times 10^{-8}$) and rs181194133 an intronic variant in *OPCML* (OR (95%CI) = 3.44 (2.24, 5.31), $p_{meta} = 2.09 \times 08^{-08}$) were all associated with increased risk of HPC in the cross-ancestral meta-analysis. Within the European ancestry, rs181777026 (11q14), located near *TENM4*, was associated with increased risk of HPC. Conversely, rs150899739 (6q24), which showed an increased risk in the Mixed ancestry but no effect in Europeans, is within *SASH1* and greatly increased the risk for HPC (OR (95% CI) = 5.84 (3.17, 10.76), $p = 1.47 \times 10^{-8}$) (Fig. S6).

At 3p21, rs1520483, an intronic variant in the lactotransferrin (*LTF*) gene, was associated with an increased risk of HPV(+) OPC (OR (95% CI) = 1.23 (1.14, 1.32), $p = 2.19 \times 10^{-8}$) in Europeans. *LTF* acts as a transcription factor, inducing expression of innate immune related genes for antiviral host defence[25,26].

rs112726671, a variant near the vitamin D receptor (VDR) gene, was associated with risk of HPV(-) OPC (OR (95%CI) = 1.23 (1.14, 1.32), $p_{meta} = 2.19 \times 10^{-8}$) in Europeans. This variant is independent from rs35189640, which is a nearby variant previously identified to increase risk of HPV(−) OPC ($r^2 = 0.0005$)[10].

**Table 1 | Summary of novel genetic variants identified in European and Mixed Groups through GWAS and Meta-Analysis**

| Population | Subsite | rsID | Mapped/Nearest Gene | CHR | BP (GRCh38) | Cytogenetic Position | Major Allele | Effect Allele | EAF (European) | EAF (Mixed) | OR (95%CI) | p-value |
|---|---|---|---|---|---|---|---|---|---|---|---|---|
| Meta-Analysis | All sites combined | rs61817953 | PIK3C2B | 1 | 204493484 | 1q32.1 | G | A | 0.188 | 0.153 | 0.90 (0.87, 0.93) | $1.17 \times 10^{-8}$ |
| | | rs6679311 | MDM4 | 1 | 204590548 | 1q32.1 | C | T | 0.285 | 0.294 | 1.11 (1.07, 1.14) | $1.25 \times 10^{-10}$ |
| | | rs1131769 | STING1 | 5 | 139478334 | 5q31.2 | C | T | 0.137 | 0.148 | 1.13 (1.09, 1.18) | $2.38 \times 10^{-09}$ |
| | | rs7334543 | BRCA2 | 13 | 32399139 | 13q13.1 | A | G | 0.272 | 0.228 | 0.91 (0.88, 0.94) | $2.39 \times 10^{-08}$ |
| | | rs7837822[a] | TP53 | 17 | 7668434 | 17p13.1 | T | G | 0.012 | - | 0.62 (0.52, 0.73) | $2.16 \times 10^{-08}$ |
| | | rs10419397 | ANKLE1 | 19 | 17280519 | 19p13.11 | G | A | 0.299 | 0.242 | 1.13 (1.10, 1.17) | $1.21 \times 14^{-14}$ |
| | OC | rs2841919 | ECSCR | 5 | 139465014 | 5q31.2 | C | T | 0.132 | 0.149 | 1.23 (1.15, 1.31) | $3.16 \times 10^{-10}$ |
| | | rs67351073 | ZGPAT | 20 | 63704213 | 20q13.33 | GA | G | 0.088 | 0.069 | 0.78 (0.72, 0.85) | $4.45 \times 10^{-08}$ |
| | | rs577454702[a] | MAPK1 | 22 | 21778123 | 22q11.22 | A | C | 0.007 | - | 2.60 (1.86, 3.65) | $2.53 \times 10^{-08}$ |
| | LA | rs55831773 | ATP1B2 | 17 | 7655719 | 17p13.1 | C | T | 0.195 | 0.219 | 1.21 (1.13, 1.29) | $5.1 \times 10^{-09}$ |
| | | rs10419397 | ANKLE1 | 19 | 17280519 | 19p13.11 | G | A | 0.295 | 0.238 | 1.18 (1.10, 1.26) | $4.33 \times 10^{-08}$ |
| | HPC | rs138707495 | GDF7 | 2 | 20677150 | 2p24.1 | T | TA | 0.009 | 0.005 | 3.06 (2.07, 4.53) | $2.33 \times 10^{-08}$ |
| | | rs77750788 | IGSF9B | 11 | 133936692 | 11q25 | G | A | 0.030 | 0.015 | 2.07 (1.61, 2.68) | $2.03 \times 10^{-08}$ |
| | | rs181194133[a] | OPCML | 11 | 132728232 | 11q25 | G | A | 0.009 | - | 3.44 (2.24, 5.31) | $2.09 \times 10^{-08}$ |
| European Ancestry GWAS | HPC | rs181777026 | TENM4 | 11 | 81037815 | 11q14.1 | C | T | 0.012 | 0.009 | 2.81 (1.94, 4.05) | $3.78 \times 10^{-08}$ |
| | HPV(−) OPC | rs112726671 | VDR | 12 | 47926100 | 12q13.11 | A | G | 0.016 | - | 2.28 (1.70, 3.07) | $4.03 \times 10^{-08}$ |
| | HPV(+) OPC | rs1520483 | LTF | 3 | 46468719 | 3p21.31 | C | T | 0.400 | - | 1.23 (1.14, 1.32) | $2.19 \times 10^{-08}$ |
| Mixed Ancestry Group GWAS | LA | rs200410709 | STXBP6 | 14 | 25417834 | 14q12 | CT | C | 0.017 | 0.013 | 3.38 (2.26, 5.07) | $3.57 \times 10^{-09}$ |
| | HPC | rs150899739 | SASH1 | 6 | 148061934 | 6q24.3 | G | A | 0.022 | 0.008 | 5.84 (3.17, 10.76) | $1.47 \times 10^{-08}$ |

Novel variants identified in meta-analysis and by group across subsites. Full list of significant variants can be found in Supplementary Data 3.

CHR Chromosome, BP Base-pair position, EAF Effect allele frequency, OR Odds ratio, OC Oral cavity, LA Larynx, HPC Hypopharynx OPC Oropharynx.

[a] These variants were only identified in European GWAS and were not present in the mixed group, as their minor allele frequency was below the 0.05% threshold determined in our analyses.

**a** **b**

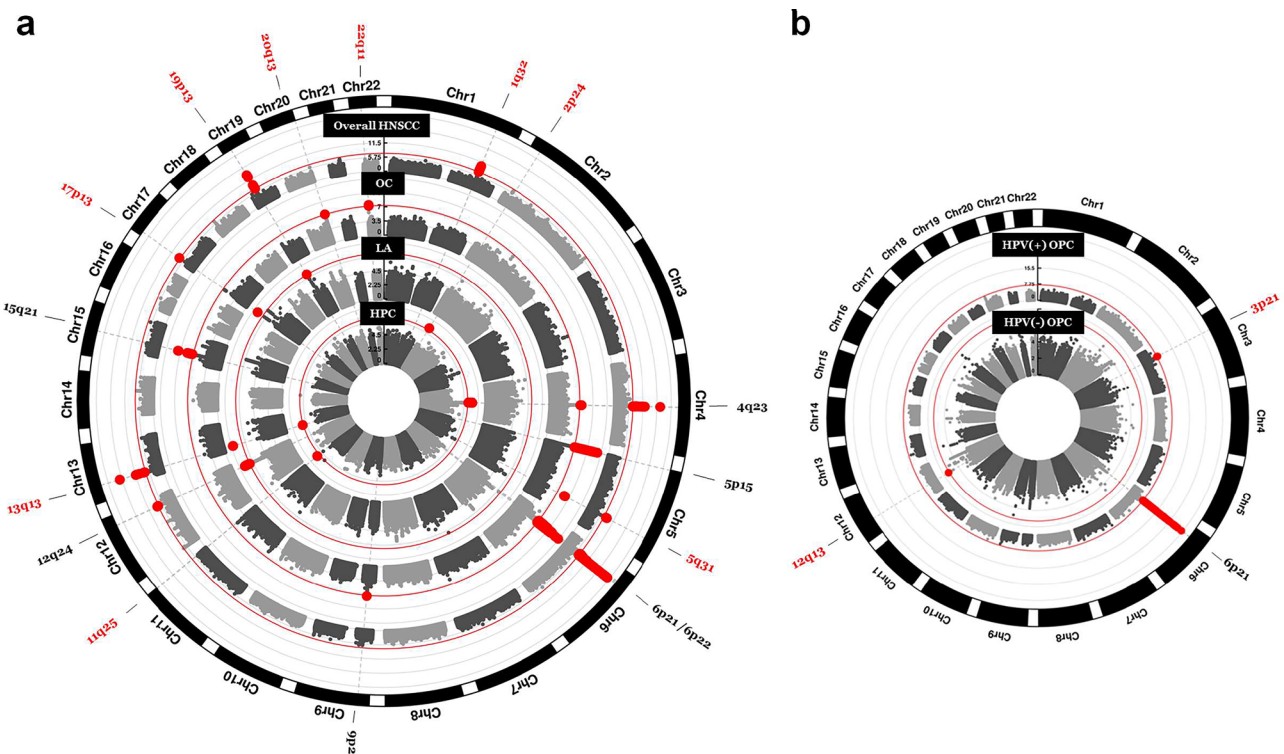

**Fig. 1 | Novel risk loci identified for HNSCC. a** Circular Manhattan plots showing novel risk loci identified in meta-analyses GWAS. Red labels indicate the cytogenetic locations of novel signals identified in meta-analyses for all sites combined or subsite-specific. Black labels represent previously identified risk loci. Red lines mark the threshold for genome-wide significance (p = 5 × 10⁻⁸). **b** Circular Manhattan plots from the European GWAS analyses of HPV(+) and HPV(−) oropharyngeal cancer. Separate Manhattan plots for each ancestry group can be found in Supplementary Fig. 1.

## Refining previously identified HNSCC risk variants

Loci identified in previous GWAS of HNSCCs at 4q23 (*ADH1B, ADH1C, ADH7*)[14], 5p15 (*CLPTM1L*)[11], 6p21 (*HLA*)[11], 6p22 (*ZNRD1-AS1*)[13], 9p21 (*CDKN2B-AS1*)[11], 12q24 (*ALDH2*)[14] and 15q21 (*FGF7*)[10] were validated here. The variants at *ADH7* (rs17529509, all sites combined) and *ADH1B* (rs1229984, oral cavity) had heterogeneity of effect by sex, where men had a significantly reduced risk compared to women (Fig. S7a, b). In colocalization analyses, we showed rs421284 near *CLPTM1L* strongly correlated with increased methylation at cg20768760 and cg21202862 and decreased methylation at cg07493874 in lung tissue, potentially implicating methylation in its mechanism of action (Supplementary Data 7).

Notably, rs11571833 (13q13), the rare (MAF: European = 0.009, Mixed = 0.007) stop gained variant, resulting in a stop codon 93 amino acids early in the BRCA2 protein, was strongly associated with an increased risk of LA (OR (95% CI) = 2.09 (1.65, 2.66), p_meta = 1.57 × 10⁻⁹) and HPC (lead variant for HPC rs11571815: OR (95% CI) = 2.73 (1.61, 3.90), p_meta = 3.99 × 10⁻⁸) separately (Fig. S7c, d). Previous GWAS combining lung and aerodigestive tract cancers, as well as studies using targeted genotyping have found this variant to substantially increase risk for smoking related cancers[27], however here to demonstrate the effect of this variant within specific subsites.

## Effect of top hits across HNSCC subsites

Subsite-specific variants described above were further evaluated to determine if these variants may be important in other subsites, but were not able to be detected due to power. Posterior probabilities of risk effects across subsites were seen for multiple variants. For variants identified as risk loci for overall HNSCC, there was evidence that specific sites likely drove the effects. For variants that were detected in a single subsite, there was evidence that these may confer also risk in other subsites (Fig. S8).

We investigated the contribution of HLA-related top hits from the GWAS to overall variance explained in risk. We found that HPV(+) OPC had the highest proportion of variability explained by this region compared to other subsites (HPV(+) OPC: 91%; HPV(−) OPC: 0%; OC: 34%; LA: 0%; HPC: 0%).

## Distinct interactions of smoking and alcohol use with risk variants

We evaluated variants for their specific interactions with smoking and alcohol use (Fig. 4). Variant effect sizes, stratified by smoking and drinking status, can be found in Figure S9. rs11571833, the *BRCA2* stop-gained variant validated here, showed clear evidence of a dose-response effect across smoking and drinking strata, but the variant did not correlate with variants related to smoking-related behaviours such as smoking initiation or cigarettes per day in colocalization analysis (Supplementary Data 7). However, the variant effect was present in both non-drinking smokers and non-smoking drinkers, suggesting the risk-increasing effect of rs11571833 requires either a carcinogenic influence. This *BRCA2* variant shows a similar gene-environment interaction separately within the European and Mixed ancestries, despite differences in sample size (Fig. 4a).

We confirm that rs1229984, the well-described missense variant in the *ADH1B* gene, has a strong protective effect on OC. which is only seen in smokers or in drinkers when stratifying by use (Figs. S7b, S9). However, we measure a strong correlation between rs1229984 and variants associated with alcoholic drinks per week but not cigarettes per day or smoking initiation (Supplementary Data 7). To separate out the linked behaviours of smoking and drinking, we investigated the association in combinations of drinking and smoking status. These analyses confirm that rs1229984 has an effect in those who smoke and drink and in non-drinking smokers but not non-smoking drinkers,

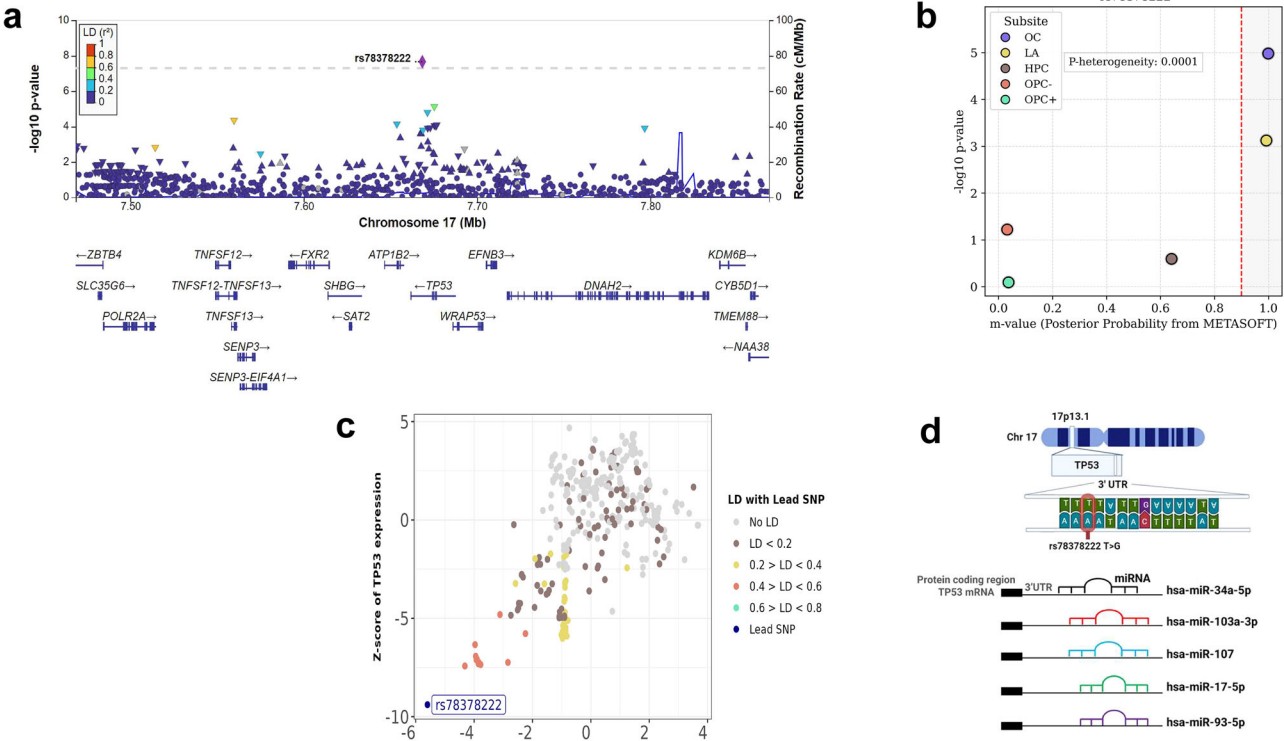

**Fig. 2 | Overview of genomic and functional characterisation of the 3' UTR variant rs78378222. a** Regional association plot for the *TP53* 3' UTR variant rs78378222 at chromosome 17p13. Each point represents a single-nucleotide polymorphism (SNP) and its association *P* value (−log₁₀ scale) from a logistic-regression test under an additive genetic model. The horizontal dashed line indicates the genome-wide significance threshold (P = 5 × 10⁻⁸). SNPs are colour-coded by their pairwise linkage disequilibrium (r²) with rs78378222, calculated in European samples from the 1000 Genomes Phase 3 reference panel (r² bins: 0–0.2, 0.2–0.4, 0.4–0.6, 0.6–0.8, 0.8–1.0). **b** PM-plot of subsite-specific association results for rs78378222. The x-axis represents the m-value, the posterior probability that a genuine genetic effect exists in each head and neck cancer subsite, estimated with METASOFT's binary-effects (BE) model. An m-value ≥ 0.9 indicates strong evidence

for an effect, ≤ 0.1 indicates no effect, and values between 0.1 and 0.9 denote uncertainty. The y-axis displays −log₁₀ P values obtained from the per-allele additive logistic-regression GWAS conducted separately for each subsite. Subsite abbreviations: OC = oral cavity, LA = larynx, HPC = hypopharynx, OPC- = HPV-negative oropharynx, OPC + = HPV-positive oropharynx. Source data are provided as a Source Data file. **c** Z-Z locus plot showing rs78378222, the lead variant, is associated with reduced *TP53* expression in whole blood, with a high PP4 score of 99%. **d** The cytogenetic location of rs78378222, along with its sequence and allele change, is mapped at the chromosomal level. According to TarBase, this variant overlaps with multiple predicted microRNA binding sites. This Figure was created in BioRender. https://BioRender.com/8c2hqe9. Source data are provided as a Source Data file.

---

suggesting the mechanism through smoking may be more important (Fig. 4b). Interactions with smoking and drinking for *ADH1C* and *ADH7* were less clear.

rs58365910 near *CHRNA5*, known to alter smoking intensity[36] showed a suggestive association with LA consistent effects across the European and Mixed ancestries (Fig. S10). The increasing risk effect of this variant was correlated with increased smoking intensity and when evaluated by exposure group, this variant shows a clear interaction with smoking but not alcohol use (Fig. 4c).

### Novel Loci in the HLA Region Specific to oral cavity and oropharynx cancer

Our genome-wide results highlight heterogeneity in the Human Leucocyte Antigen (HLA) region, which encodes genes involved in immune response, across HNSCC subsites. For HPV(+) OPC, signals were identified at both 6p21 and 6p22, but for OC, only the 6p21 signal was seen. The HLA region is particularly susceptible to genetic diversity across populations and is highly polymorphic with a dense LD structure. To account for this, genotyped variants in this region were re-imputed to an HLA-specific reference to define variants, amino acid changes and 4-digit alleles, which were then analysed separately using fine mapping strategies to identify independent signals. Independence of signals was carefully evaluated using linkage and conditional analysis (Supplementary Data 8, 9).

Overall, 19 independent signals reached significance (Supplementary Data 10, 11). Eleven risk variants were identified specific to OC, HPV(+) OPC, HPV(−) OPC, and for HNSCC overall (Table 2 and Fig. S11). Novel variants were defined as being both independent from lead variants reported across subsites and from previously reported variants (Supplementary Data 8).

Three intronic variants were associated with the risk of HNSCC overall. The Chr6:33046667 variant, near *HLA-DPB1* (OR (95% CI) = 1.11 (1.07, 1.14), $p_{meta} = 1.32 \times 10^{-8}$) and rs28360051 near *PSORS1C3* (OR (95% CI) = 1.23 (1.14, 1.34), $p_{meta} = 1.91 \times 10^{-7}$) both increased HNSCC risk in the cross ancestral meta-analysis. The rs28360051 variant was strongly driven by its effect in HPV(+) OPC (Fig. S11a). An intronic variant, rs1536036, mapping to *ITPR3*, a receptor that mediates the release of intracellular calcium, was protective for HNSCC overall (OR (95% CI) = 0.85 (0.80, 0.91), p = 8.42 × 10⁻⁷) only in the admixed ancestry.

For HPV(+) OPC, five variants were identified. rs4143334, in the noncoding transcript exon of *ZDHHC20P2* increased cancer risk (OR (95% CI) = 1.89 (1.51, 2.35), $p_{meta} = 1.91 \times 10^{-8}$). The remaining three cross ancestral variants had important functional significance. The first (DRB1 37Asn/Ser) causes an amino acid change in the antigen-binding pocket (P9 pocket) of the beta chain of the HLA-DR protein and reduces HPV(+) OPC risk (OR (95% CI) = 0.68 (0.63, 0.73), $p_{meta} = 3.22 \times 10^{-23}$). The second (HLA-B 67Cys/Ser/Tyr) is in an antigen

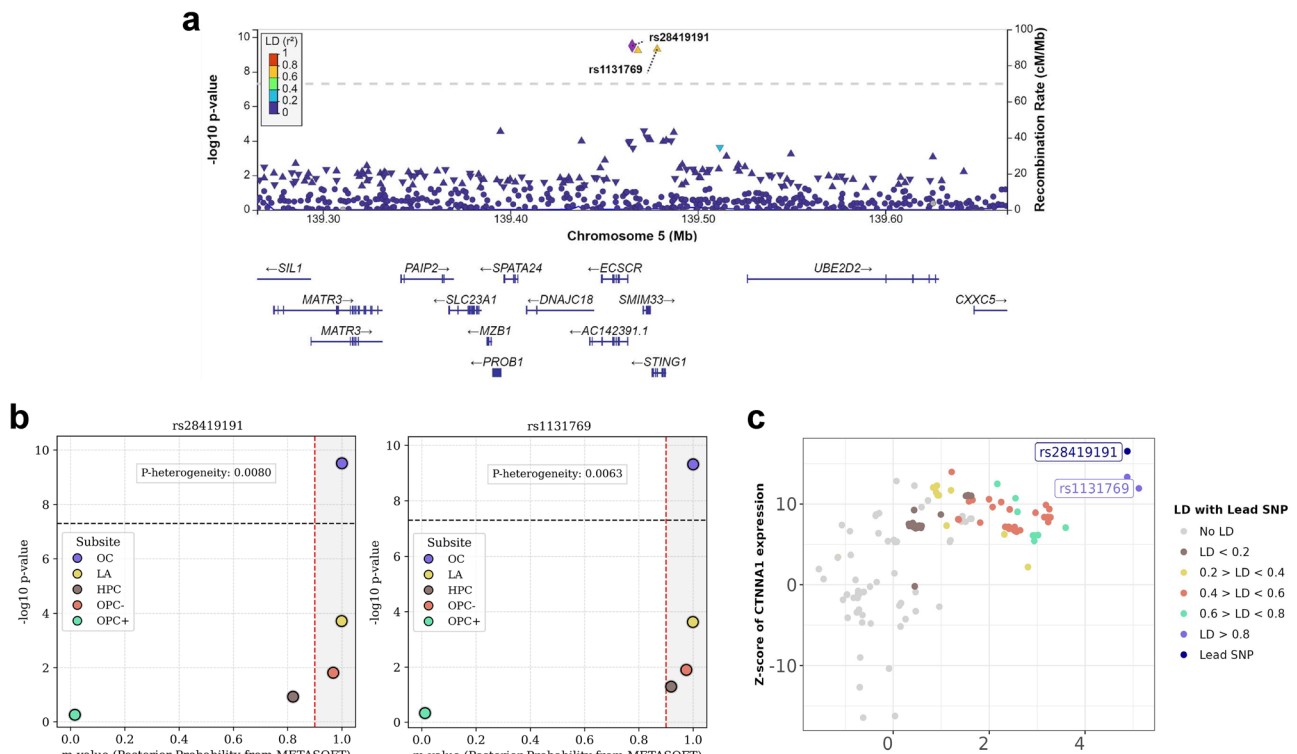

**Fig. 3 | Genomic and functional characterisation of 5q31 variants rs28419191 and rs1131769. a** Regional association plot for the two independent lead single-nucleotide polymorphisms (SNPs) rs28419191 and rs1131769 on chromosome 5. Each point shows the $-\log_{10} P$-value from a logistic-regression test under an additive genetic model in each cohort. The horizontal dashed line marks the genome-wide significance threshold ($P = 5 \times 10^{-8}$). **b** PM-plots for rs28419191 and rs1131769. On the plot, the x-axis represents the m-value, the posterior probability that a genuine genetic effect exists in each head and neck cancer subsite, estimated with METASOFT's binary-effects (BE) model. An $m$-value $\geq 0.9$ indicates strong evidence for an effect, $\leq 0.1$ indicates no effect, and values between 0.1 and 0.9 denote uncertainty. The y-axis displays $-\log_{10} P$-values obtained from the per-allele additive logistic-regression GWAS conducted separately for each subsite. Subsite abbreviations: OC = ral cavity, LA = larynx, HPC = hypopharynx, OPC- = HPV-negative oropharynx, OPC + = HPV-positive oropharynx. Source data are provided as a Source Data file. **c** Z-Z locus plot showing colocalization of rs28419191 and rs1131769 with *CTNNA1* expression in whole blood, both with a PP4 score of 99%.

binding pocket (B-pocket) of HLA-B and also results in decreased HPV(+) OPC risk (OR (95% CI) = 0.81 (0.74, 0.88), $p_{meta} = 1.33 \times 10^{-6}$) (Fig. 5a). The third (DRB1 233Thr), is in exon 5 of *DRB1* and increases risk of HPV(+) OPC (OR (95% CI) = 1.27 (1.17, 1.38), $p_{meta} = 7.15 \times 10^{-9}$). This amino acid change is in high LD with several others that are in the HLA-DR binding pocket, of which 5 have similar risk (Supplementary Data 12). Accuracy of best-fit models, which included each related amino acid in place of DRB1 233Thr, were found to be similar to the original model containing DRB1 233Thr ($\triangle$BIC $\pm$ 2), indicating that presence of any of these five amino acid changes—including DRB1 10Glu/Gln and 12Lys located in the HLA binding pocket—confers similar levels of risk (Fig. 5b and Supplementary Data 12). Within those of European ancestry, the HLA-B*51:01 allele increased the risk of HPV(+) OPC (OR (95% CI) = 1.9 (1.55, 2.31), $p_{meta} = 3.6 \times 10^{-10}$).

For HPV(-) OPC, rs1131212 was found to be associated with an increased risk cross ancestrally (OR (95%CI) = 1.33 (1.19, 1.49), $p_{meta} = 5.33 \times 10^{-7}$) (Fig. 6a). This functional variant maps to exon 2 of the *HLA-B* gene, causing an amino acid change Gln94His in an *HLA-B* binding pocket. rs1131212 tags another functional HLA-B amino acid change, HLA-B 70Asn/Ser (Supplementary Data 10) in strong LD ($r^2 = 1$), which has a similar effect and with similar model accuracy (OR (95% CI) = 1.32 (1.18, 1.47), $p_{meta} = 8.81 \times 10^{-7}$) (Fig. 6b and Supplementary Data 12). These results suggest that the presence of either rs1131212 or HLA-B 70Asn/Ser is equivocal to increase cancer risk across ancestries.

The HLA-A*24 allele tagged the known intronic variant rs1264813 in *MICD*, and was similarly associated with increased risk of HPV(-) OPC (OR (95% CI) = 1.34 (1.18, 1.52), $p_{meta} = 7.24 \times 10^{-6}$) cross ancestrally. Accuracy of the model including this allele was similar to the model

including rs1264813, suggesting these signals convey similar risk (Fig. 6c and Supplementary Data 12).

A haplotype was identified that tagged the known intronic variant, rs9268925 in *DRB9*, and was associated with decreased risk of OC (OR (95% CI) = 0.8 (0.73, 0.86), $p_{meta} = 2.15 \times 10^{-8}$). The haplotype, DRB1*15:01-DQA1*01:02-DQB1*06:02, had a similar risk and similar model accuracy compared to the known variant, suggesting that this variant and the haplotype can be used interchangeably to measure this risk across ancestries (Fig. 6d and Supplementary Data 12). Two variants specific to the European ancestry were associated with the risk of OC: DRB1 74Ala/Leu/Del (OR (95% CI) = 0.82 (0.77, 0.87), $p = 4.94 \times 10^{-10}$) and rs9267280 (OR (95% CI) = 1.32 (1.19, 1.47), $p = 3.48 \times 10^{-7}$).

**Cross ancestry equivalent of established risk variants, including the well-known haplotype DRB1*13:01-DQA1*01:03-DQB1*06:03**
The DRB1*13:01-DQA1*01:03-DQB1*06:03 haplotype is well known to reduce the risk of cervical cancer and HPV(+) OPC[10,11,28]. Notably, the two *DRB1* amino acid changes, DRB1 37Asn/Ser and DRB1 233Thr, described here for risk of HPV(+) OPCs are within this haplotype (Fig. 5a). To determine if the haplotype is completely represented by these amino acid changes, we replaced the amino acids with the full haplotype in the risk model for HPV(+) OPC (Fig. 5c). Unexpectedly, the effect of HLA-B 67Cys/Ser/Tyr disappeared when including the haplotype, suggesting these are shared risk loci. When all three variants were replaced by the haplotype, the haplotype was independently associated with HPV(+) OPC risk (OR (95% CI) = 0.53 (0.43, 0.63), $p_{meta} = 1.76 \times 10^{-10}$), as described previously[11]. Importantly,

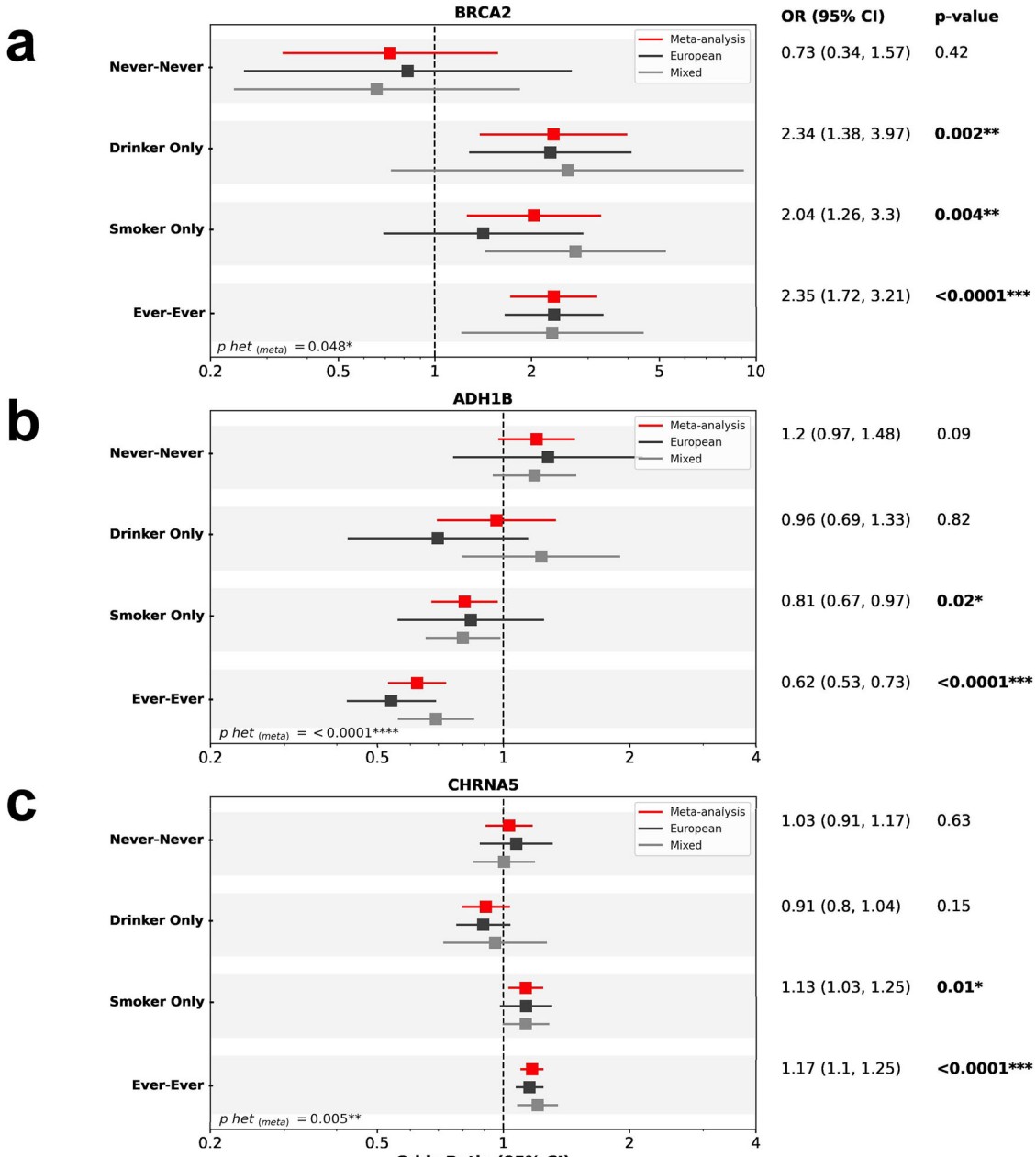

**Fig. 4 | Gene-environment interactions with alcohol and smoking.** Effect estimates for (**a**) rs11571833 (*BRCA2*) (**b**) rs1229984 (*ADH1B*), and (**c**) rs58365910 (*CHRNA5*) stratified by smoking and drinking (Never smoker-Never drinker, Smoker Only, Drinker Only, and Ever smoker-Ever drinker) from the meta-analysis, and within European and Mixed groups. In each panel, Odds ratios and CIs were estimated by logistic regression under an additive genetic model in each ancestral group and then combined by fixed-effects inverse-variance meta-analysis. For each exposure/genotype category, we report the exact number of independent subjects: never-smoker/never-drinker ($n = 876$ cases, 2713 controls), drinker only ($n = 726$ cases, 9552 controls), smoker only ($n = 2739$ cases, 2242 controls) and ever-ever (both smoker and drinker; $n = 4860$ cases, 10,002 controls). heterogeneity among the 2 ancestries assessed by Cochran's Q test. Only the odds ratios (OR) and 95% confidence intervals (CI), p-value and p-heterogeneity (p-het) for the meta-analysis are shown here. Source data are provided as a Source Data file.

however, model accuracy was highest for the model consisting of the original three amino acid changes compared to the haplotype, suggesting that the specific independent effects of the newly identified DRB1 37Asn/Ser, DRB1 233Thr, and possibly HLA-B 67Cys/Ser/Tyr underlie the effect of the DRB1*13:01-DQA1*01:03-DQB1*06:03 haplotype. The importance of these amino acid changes is highlighted by their allele frequencies across populations, compared to the haplotype (Fig. 5d). The allele frequency of the haplotype across genetic ancestries is low and ranges from 3% to 6%, while the frequency of the three amino acids across ancestries is much higher, ranging from 26% to 33%.

The rs2523679 variant, which decreases risk of HPV( + ) OPC (OR (95% CI) = 0.63 (0.53, 0.75), $p_{meta} = 2.26 \times 10^{-7}$), tags the established HLA-B*15:01 ($r^2 = 0.51$) and HLA-B 156Trp ($r^2 = 0.51$) signals that were previously found in those of European ancestry. Here we show that while the effects of HLA-B*15:01 and HLA-B 156Trp remain specific to European ancestry, rs2523679 confers a similar level of risk for both European and admixed populations, providing a cross-ancestral equivalent of this loci (Fig. 5e). Other cross-ancestral validated loci are described in Supplementary Data 11 and 12.

**Table 2 | Summary of Novel Genetic Variants Identified Across Ancestry-Specific and Meta-Analysis of HLA-fine mapping in all sites combined and subsite-specific**

| Population | Subsite | Variant | Gene | Locus; Cytoband | Impact | Ref[a] | Effect Allele[a] | OR (95% CI) | p-value[b] |
|---|---|---|---|---|---|---|---|---|---|
| Meta-Analysis | All sites combined | Chr6:33046667 | HLA-DPB1 | class II | intron | C | T | 1.11 (1.07,1.14) | 1.32 × 10⁻⁸ |
| | | rs28360051 | PSORS1C3 | | intron | G | A | 1.23 (1.14,1.34) | 1.91 × 10⁻⁷ |
| | HPV(-) OPC | rs1131212 | HLA-B | class I | Gln94His | C | G | 1.33 (1.19,1.49) | 5.33 × 10⁻⁷ |
| | HPV(+) OPC | DRB1 37Asn/Ser | HLA-DRB1 | class II | Amino acid change | Ab | Pr | 0.68 (0.63,0.73) | 3.22 × 10⁻²³ |
| | | rs4143334 | ZDHHC2OP2 | class I | Non-coding transcript exon | A | G | 1.89 (1.51,2.35) | 1.91 × 10⁻⁸ |
| | | DRB1 233Thr | HLA-DRB1 | class II | Amino acid change | Ab | Pr | 1.27 (1.17,1.38) | 7.15 × 10⁻⁹ |
| | | B 67Cys/Ser/Tyr | HLA-B | class I | Amino acid change | Ab | Pr | 0.81 (0.74,0.88) | 1.33 × 10⁻⁰⁶ |
| Admixed | All sites combined | rs1536036 | ITPR3 | | Intron | A | G | 0.85 (0.80,0.91) | 8.42 × 10⁻⁷ |
| European | OC | DRB1 74Ala/Leu/Del | HLA-DRB1 | class II | Amino acid change | Ab | Pr | 0.82 (0.77,0.87) | 4.94 × 10⁻¹⁰ |
| | | rs9267280 | MICB-DT | class I | Intron | G | A | 1.32 (1.19,1.47) | 3.48 × 10⁻⁷ |
| | HPV(+) OPC | HLA-B*51:01 | HLA-B | class I | Amino acid change | Ab | Pr | 1.9 (1.55,2.31) | 3.6 × 10⁻¹⁰ |

Novel variants identified in the meta-analysis and ancestry-specific groups are shown here. No novel variants were identified within Middle Eastern, African and South Asian populations. Full list of variants in this region can be found in Supplementary Data 10. All variants were tested for independence, which was defined by linkage disequilibrium (R2) > 0.3 and a Bonferroni threshold of P > 10-6 when conditioning on variants from other subsites or with previously identified variants. Further details can be found in Supplementary Data 8 and 9.

[a]Ref/A1 allele is in binary marker format (Ab = Absent, Pr = Present) of classical HLA alleles, amino acid residue, HLA intragenic, insertions/deletions. (see: https://imputationserver.readthedocs.io/en/latest/pipeline).

[b]meta-analysis P-value of the final model including all significant independent variants adjusted by biological sex, imputation batch and PCs.

HLA significance level = 2.4 × 10-6 considering all variants in chr6.

## Discussion

Across the GWAS and HLA-focused analyses, we identify 18 genome-wide and 11 HLA-specific novel variants associated with the risk of HNSCC. Due to increased power compared to previous GWA studies, we identified genetic variants including in *TP53 and STING1* and validated known variants in *BRCA2* separately in LA and HPC, two under-studied cancer sites, as well as multiple signals in HPC, such as *GDF7*. Variants from fine mapping highlight key differences in HLA associations between HPV(+) OPCs, HPV(-) OPCs and OCs. Post-GWAS analyses, including colocalization and the use of harmonised individual-level risk factor data, enabled the investigation of variant function and variant-environment interactions.

A key finding was the identification of the low-frequency rs78378222 variant located in the 3′ UTR of *TP53* with a protective effect against overall HNSCC. This variant modulates *TP53* gene regulation, at transcriptional and post-transcriptional levels as indicated by eQTL and sQTL analyses, with decreased *TP53* expression correlating with a reduced risk of overall HNSCC. This finding supports a previous candidate SNP study in a non-Hispanic white population assessing its effect on HNSCCs (OR = 0.44, 95% CI: 0.24,0.79, p = 0.008)[29]. Interestingly, while this variant is protective for HNSCCs and breast cancers[30], it increases the risk of skin basal cell carcinoma[21], brain tumours[21], colorectal adenocarcinoma[21], oesophageal SCC[31], prostate cancers[21], and neuroblastoma[32]. While these findings suggest a tissue-specific regulatory impact, the precise effects of rs78378222 in head and neck tissues remain to be elucidated. Further functional studies are necessary to determine how this variant influences *TP53* expression and splicing in head and neck tissues. Furthermore, how these alterations may affect p53's tumour suppressive activities and somatic mutations in the context of head and neck cancer are important areas for future study.

Two closely linked genetic variants were identified in 5q31, including the missense variant rs1131769 found in the cyclic dinucleotide (CDN) binding domain of the *TMEM173* gene, of which the resultant STING1 protein detects viral DNA and bacterial CDNs to activate the host immune response in humans. Notably, this variant shows no association with HPV( + ) OPC, but a consistent increased risk for all non-HPV cancer types. Both variants also showed evidence of eQTLs for *CTNNA1*, a gene in which germline genetic variants are known to cause Hereditary Diffuse Gastric Cancer[33].

We were able to validate several known HNSCC risk variants and further investigate their interaction with major risk factors. rs11571833 has been linked with lung and upper aerodigestive tract cancers[27]; here we demonstrate that this effect is largest in LA and HPC cancers. This variant, found in *BRCA2*, causes a 93 amino-acid deletion including the RAD51 binding domain, important in the Fanconi Anaemia Pathway for double-strand DNA repair, and is distinct from the highly penetrant familial *BRCA* mutations[34]. Previous literature suggests smoking is mainly implicated in the mechanism of action of rs11571833[27]. However, here we provide evidence across ancestry and separately in the European and Mixed ancestry groups that this variant increases HPV-negative cancer risk with either the exposure of smoking or drinking, and that there is no effect in never-smoking non-drinkers. This supports the theory that DNA repair to environmental factors is disrupted[34] and suggests that the crucial DNA damage in HNSCC can be contributed to by alcohol use or smoking. In similar analyses, we show the well-known *ADH1B* variant rs1229984 confers a protective effect for OC, which is strongest in non-drinking smokers, suggesting a mechanism through smoking as well as the established one through alcohol use. The *CHRNA5* variant, rs58365910, was identified as a suggestive association for risk of LA cancer. As expected, this variant only shows an effect in smokers, suggesting that it acts through its known effect on smoking heaviness, a phenotype defined by cigarettes smoked per day[35]. These variants show specific interactions with smoking/drinking; future studies could investigate polygenic risk

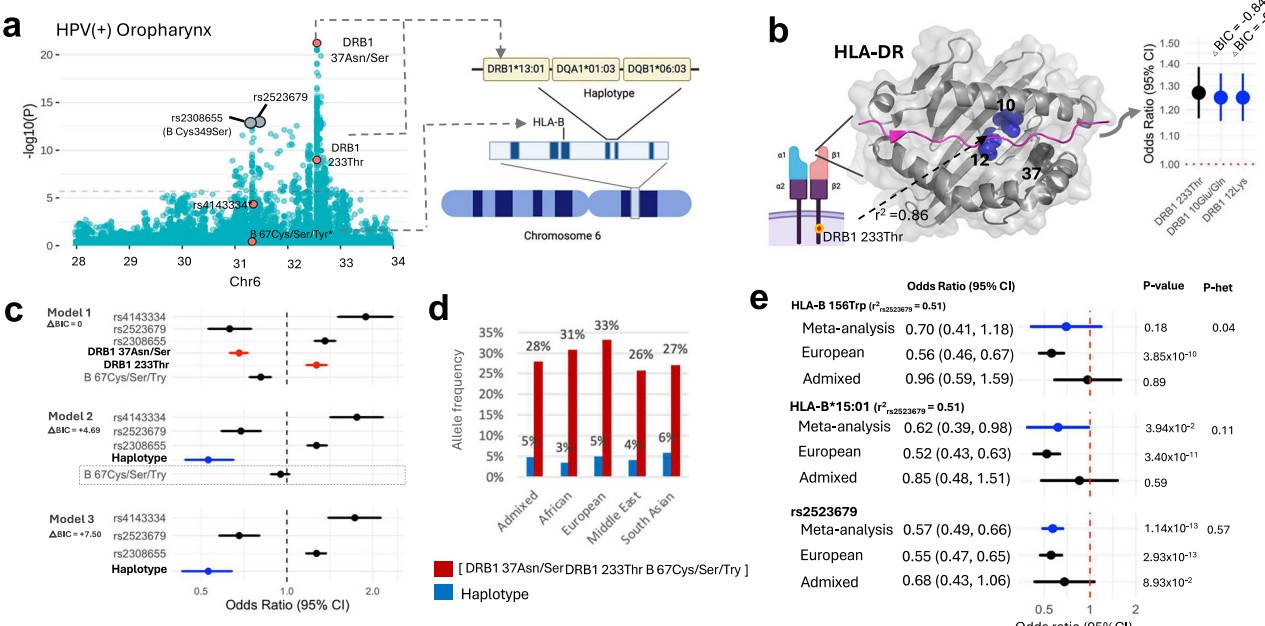

**Fig. 5 | Cross-ancestry HLA risk loci of HPV(+) OPC. a** Manhattan plots showing all independent lead variants for risk of HPV(+) OPC (cases=2,207; controls=38,973). Variants highlighted under the significance threshold reached significance in later rounds; only the plot from the first round of stepwise logistic-regression analysis is shown here. Novel variants are orange; known variants are grey. The horizontal red line reflects the HLA significance threshold ($p < 2.4 \times 10^{-6}$), adjusted using the Bonferroni correction. DRB1 37Asn/Ser, DRB1 233Thr, are within DRB1*13:01-DQA1*01:03-DQB1*06:03, while HLA-B67Cys/Ser/Try was associated with the haplotype. **b** Out of the five interchangeable amino acid residues in LD with DRB1 233Thr (OR = 1.27, 95% CI:1.17,1.38, $p = 7.15 \times 10^{-9}$), with △BIC ± 2, DRB1 12Lys (OR = 1.25, 95% CI:1.16,1.35, $p = 1.97 \times 10^{-8}$), and DRB1 10Glu/Gln (OR = 1.25, 95% CI:1.16,1.35, $p = 1.56 \times 10^{-8}$) are in the HLA-DR binding pocket and have similar effects. **c** Model accuracy and risk estimates of amino acid residues and haplotypes. Model 1: six variants identified from fine-mapping, DRB1 233Thr (OR = 1.27, 95% CI:

1.17,1.38, $p = 7.15 \times 10^{-9}$), DRB1 37Asn/Ser (OR = 0.68, 95% CI: 0.63,0.73, $p = 3.22 \times 10^{-23}$), rs2523679 (OR = 0.63, 95% CI: 0.53,0.75, $p = 2.26 \times 10^{-7}$), rs4143334 (OR = 1.89, 95% CI: 1.51,2.35, $p = 1.91 \times 10^{-8}$), rs2308655 (OR = 1.36, 95% CI: 1.25,1.48, $p = 3.91 \times 10^{-12}$), and HLA-B 67Cys/Ser/Tyr (OR = 0.81, 95% CI: 0.74,0.88, $p = 1.33 \times 10^{-6}$), used as the baseline reference model; Model 2: replaces DRB1 37Asn/Ser, DRB1 233Thr (highlighted in red) with the haplotype (OR = 0.53, 95% CI: 0.44,0.65, $p = 3.82 \times 10^{-10}$) (highlighted in blue), effect of HLA-B 67Cys/Ser/Try disappears (OR = 0.95, 95% CI: 0.88,1.02, $p = 0.16$); Model 3: All 3 amino acids replaced with haplotype (OR = 0.53, 95% CI: 0.44,0.65, $p = 3.82 \times 10^{-10}$). **d** Allele frequencies of DRB1*13:01-DQA1*01:03-DQB1*06:03 and of having all three amino acid residues by ancestry. **e** The HLA-B 156Trp amino acid change and the HLA-B 15:01 allele are specific to European ancestry, but the rs2523679 variant, which is in LD with both, has a cross-ancestral effect. His figure is created in BioRender. https://BioRender.com/98q9ivz. Source data are provided as a Source Data file.

scores within these strata to aid personalised prevention strategies. Additionally, while we focused primarily on tobacco and alcohol exposures, there are emerging risk factors such as air pollution, aging and poor oral hygiene that should be explored in future studies.

Through HLA fine mapping efforts, we identified 11 loci specific to the HLA region, of which eight were separately associated with risk of OC, HPV(+) OPC or HPV(-) OPC. Most of the class I loci were found in *HLA-B*, while most of the class II loci were in *DRB1*. Given the dense, overlapping structure of the HLA region, we also identified functionally equivalent signals at the amino acid, allele, or haplotype level, enabling these data to support a variety of downstream applications requiring functional information.

A previously unreported class II haplotype was identified for risk of OC, DRB1*15:01-DQA1*01:02-DQB1*06:02. This haplotype has been found to reduce autoantibody development and abnormalities of metabolic traits, such as dysglycemia. As such, this haplotype was found to be protective against progression of type I diabetes (DM)[36]. The relevance of this finding is evidenced by a meta-analysis that found that individuals with DM have a higher risk of developing oral cancer[37], potentially related to DM-related metabolic traits such as hypertension and dyslipidemia[38]. Nevertheless, a link between DM and OC remains inconsistent[39–41]. The OC-specific validated variant, rs4990036, is also associated with a non-HPV infection, varicella zoster[42], highlighting that other infections may be important in cancer risk. This is especially important considering the oral microbiome as a potential emerging risk factor for oral cavity cancer[43,44].

The well-known haplotype, DRB1*13:01-DQA1*01:03-DQB1*06:03, has been found to be protective against cervical cancer and HPV(+) OPC, highlighting its role in detecting HPV infection[10,11,28]. This haplotype is present at about 5% in the European ancestry and is less common in other ancestries. We show here that the DRB1*13:01-DQA1*01:03-DQB1*06:03 haplotype is represented by the three amino acid changes identified in this work, DRB1 37Asn/Ser, DRB1 233Thr, and HLA-B 67Cys/Ser/Tyr. Notably, however, the amino acid changes themselves more precisely estimate the risk of HPV(+) OPC across ancestries and likely drive the effect of the haplotype across ancestries. The higher allele frequencies of the amino acids, ranging from 26% to 33%, allow for better detection of subjects with increased risk for HPV(+) OPC across populations.

The intronic rs2523679 variant is a cross-ancestral equivalent of HLA-B*15:01 and HLA-B 156Trp, two previously identified European-specific variants. This now identified variant can be used to evaluate risk of HPV(+) OPC across multiple ancestries, and highlights the importance of including non-European populations, even with limited sample size.

In this work, we were limited by the power of non-European populations, forcing us to combine multiple populations. Although this did provide additional power for discovery, it will have reduced the ability to identify variants specific to certain populations. Where variants were specific to non-European ancestries, we were able to assess these in the different populations, but increased sample sizes from more diverse populations should still be seen as a priority in this field.

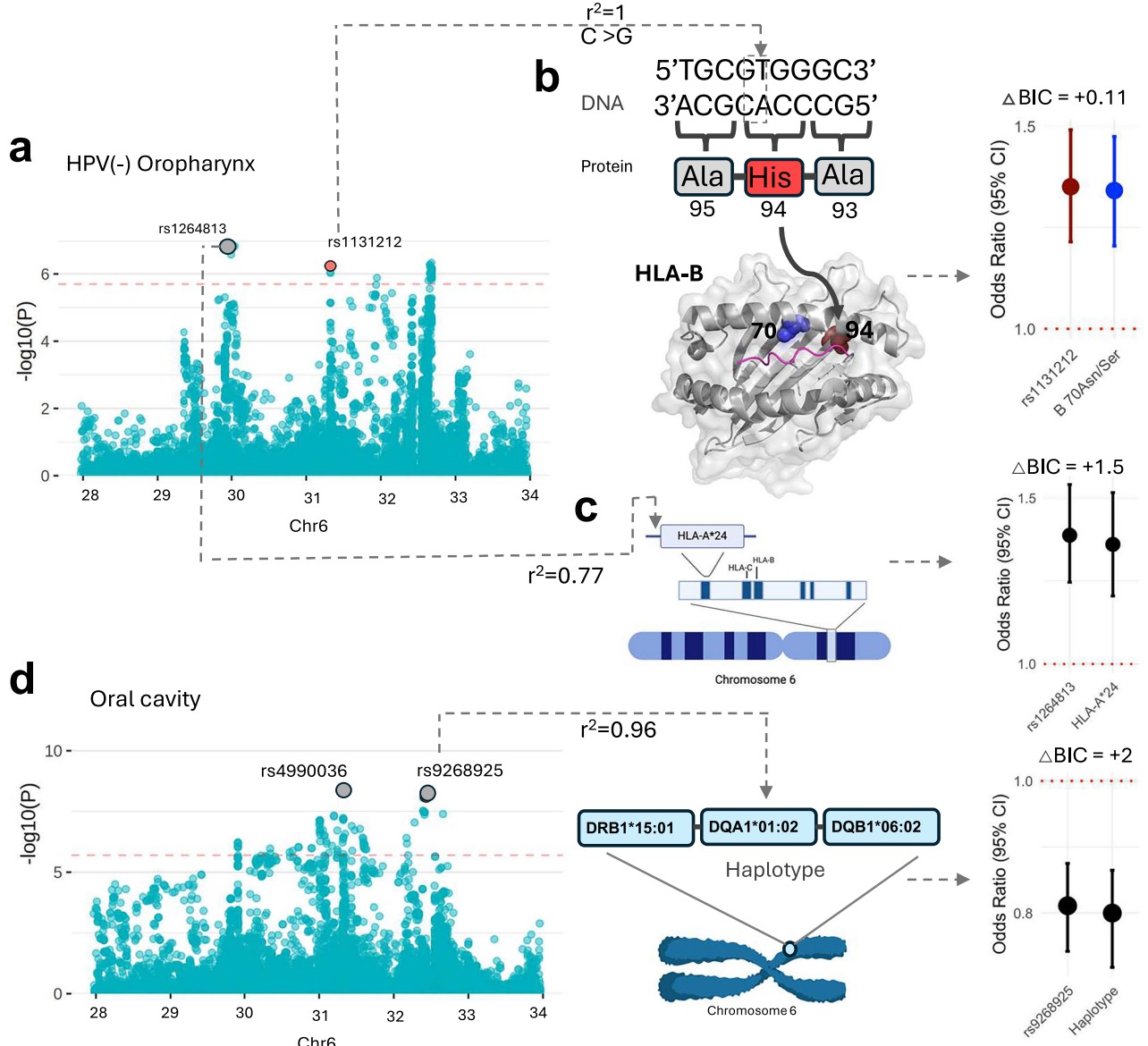

**Fig. 6 | Novel HLA risk loci for HPV(-) oropharynx and oral cavity cancer.**
Manhattan plots display all independent lead variants of risk for HPV(−)(cases =
1470; controls = 38,973) and OC (cases = 5578; controls =38,973) subsite. Variants
highlighted under the significance threshold reached significance in later rounds;
only the plot from the first round of stepwise logistic-regression analysis is shown
here. Novel variants are highlighted in red; known variants are in grey. The hor-
izontal red line reflects the HLA significance threshold (p < 2.4 × 10⁻⁶), adjusted
using the Bonferroni correction. **a** HPV(-) oropharynx: The lead SNP, (**b**) rs1131212
(OR = 1.33, 95% CI:1.19,1.49, p = 5.33 × 10⁻⁷), causes an amino acid change from Gln
to His at residue 94 located in the HLA-B protein binding pocket (PDB ID: 2BVP).
This variant is in LD (r² = 1) with 70Asn/Ser (OR = 1.32, 95% CI:1.18,1.47,
p = 8.81 × 10⁻⁷). The right panel shows the comparable risk effects of the two related

signals. The known SNP, (**c**) rs1264813 (OR = 1.37, 95% CI:1.22,1.55, p = 2.77 × 10⁻⁷), is
in high LD (r² = 0.77) with HLA-A*24 allele (OR = 1.34, 95% CI:1.18,1.52, p = 7.24 × 10⁻⁶)
and shows comparable risk effects shown in right panel. d) Oral cavity: The lead
SNP, rs9268925 (OR = 0.81, 95% CI: 0.75,0.87, p = 1.36 × 10⁻⁷), is highly correlated
with a novel risk haplotype, DRB1*15:01-DQA1*01:02-DQB1*06:02 (OR = 0.8, 95%
CI:0.73,0.86, p = 2.15 × 10⁻⁸), and has a similar risk effect, as shown in the right panel.
Model accuracy difference (△BIC) between the original model in the presence of
all independent lead variants and the model replacing the lead variant with a related
amino acid residue, allele or haplotype, lower than 2 confer equivalent risk. This
figure is created in BioRender. https://BioRender.com/98q9ivz. Source data are
provided as a Source Data file.

Although analysing all-site HNSCC can be beneficial, it must be
remembered that these cancers are heterogeneous, and the subsite ana-
lyses provide a clearer picture of the genetic architecture of the condi-
tions. Where we identify genetic variants in one site, we assess the effect of
this variant across all subsites to assess the heterogeneity, but despite the
increased sample size in this study, there may still be limited power for
discovery, especially in the less common subsites such as HPC. In addition,
future studies using the variants identified here for downstream analysis
should consider performing further validations, particularly for those with
borderline genome-wide significance.

In summary, in this HNSCC GWAS, which includes diverse popu-
lations, we identify 18 novel genome-wide associated variants and 11
HLA-specific novel variants associated with HNSCC and its subsites,
including rs78378222 in the *TP53* 3′ UTR, which confers a 40% reduction
in odds of developing overall HNSCC. We expand knowledge of the
gene-environment relationship of *BRCA2* and *ADH1B* variants, demon-
strating that their effects act through both smoking and alcohol use.
Finally, a focus on the HLA region highlighted that although HPV( + )
OPC, HPV(-) OPC and OC all show GWAS signal at 6p21, each subsite has
distinct associations at the variant, amino acid and 4-digit allele level.

## Methods

This research complies with all relevant ethical regulations. All contributing studies obtained ethics approval from their respective local Institutional Review Boards (IRBs) or ethics committees, with written informed consent from participants. For the meta-analysis and secondary analysis of individual-level data, ethics approval was granted by the International Agency for Research on Cancer (IARC) Ethics Committee (IEC 19–38).

### Study design and populations

Individual-level data came from 18 studies across 23 countries in Europe, the Middle East, North America, South America, and South Asia, and 9 genotyping arrays (Supplementary Data 1). Data on demographics (sex, age, country), diagnosis (TNM status, year of diagnosis, ICD code −7th edition), HPV status (HPV16E6 serology, P16 immuno-histochemistry (IHC), and HPV DNA in situ hybridisation (ISH)) and self-reported behaviours (smoking status, packyears, and drinking status) were collated and harmonised across all study participants. Sex was genetically determined using genotype data; this sex variable was used throughout the study to account for potential genetic effects of sex. Eligible sites for inclusion consisted of the oral cavity (C00.3, C00.4, C00.5, C00.6, C00.8, C00.9, C02.0–C02.9 (except C02.4 and C02.8), C03.0–C03.9, C04.0–C04.9, C05.0–C06 (except C05.1, C05.2)); oropharynx (C01-C01.9, C02.4, C05.1, C05.2, and C09.0–C10.9); hypopharynx (C12.0-C13.0); larynx (C32); and unknown primary site/overlapping/not otherwise specified (NOS) sites (C14, C05.8, C02.8, C76.0). Base of tongue (C01) and tonsils (C09) were grouped with oropharynx, as these sites are frequently driven by HPV16. For studies with available information on HPV infection for OPC tumours, the HPV status provided by the centre was used (P16 status, HPV DNA ISH, or HPV serology). When information from various methods was available, a positive HPV status was determined by the presence of the HPV16 E6 antibody in serology. If serology data were absent, dual positivity of p16 and HPV DNA ISH was classified as HPV positive (HPV(+)), while dual negativity of p16 and HPV DNA ISH was classified as HPV negative (HPV(-)). Any other combinations of test results were considered as "not available"[45].

Nineteen studies were included here with either multi-centre case-control, cohort, or clinical trial study designs. Previously generated data were either downloaded from dbGap, requested through controlled access from relevant consortia, or contributed by the study PIE, who contributed 10,404 cases and 34,596 controls. Controls from the UKBiobank study were selected by first excluding UKBiobank participants with a previous cancer diagnosis or missing data on key variables such as smoking or alcohol use, and then randomly selecting 10 controls for each HNSCC case. New genotyping data were generated for 8,669 cases and 4,261 controls and were not included in any previous GWAS. All study details, including data sources, dbGap accession numbers and case control distributions across subsites, can be found in Supplementary Data 1. Power calculations for the analyses can be found in Supplementary Data 13.

### Genotype quality control and imputation

A flow diagram detailing the preparation of the genetic data can be found in the supplementary material (Fig. S12). Genotypes were generated from nine different genotyping arrays (Supplementary Data 1). All newly generated genotype data were called using GenomeStudio (Illumina, 2014). Quality control steps were conducted within each array. All genotype data were converted to genome build 38, using the LiftOver programme (https://genome.ucsc.edu/cgi-bin/hgLiftOver) to convert from previous builds. Genotype data was checked and corrected for consistency of strand, positions and reference alleles. Quality control was conducted using the PlinkQC package[46] in R, utilising PLINK 1.9[47]. Samples were filtered for sex mismatch (males with SNP sex <0.8; females with SNP sex >0.2), missingness (>3%),

heterozygosity (>3 standard deviations from the mean) and cryptic relatedness (identity-by-descent > 0.185). Variants were filtered for genotype missingness (>1%), deviation from Hardy Weinberg equilibrium ($p < 1 \times 10^{-5}$) and minor allele count (<20). The number of samples and variants removed at each QC step is provided in Supplementary Data 2 and Supplementary Data 14. All arrays were imputed to the TOPMED imputation panel[48] separately using the TOPMED Imputation server[49].

To increase the number of controls comparable to the participants in the HN5000 study, 17,815 additional participants (including known related individuals) were included from the Avon Longitudinal Study of Parents and Children (ALSPAC), which had been previously genotyped (Supplementary Data 1)[50,51]. To account for potential batch effects between the HN5000 study (Infinium Global Screening Array [GSA]) and additional ALSPAC controls (Illumina 550 Quad, Illumina 660 W Quad), a double imputation approach was applied (Supplementary Note 1). Briefly, GSA HN5000 cases and the additional controls were imputed to the TOPMED reference panel separately as detailed above. Following this step, variants which were (i) genotyped on both arrays, (ii) genotyped on the GSA with high-quality imputation ($R^2$ score >0.9) on the ALSPAC array, and (iii) genotyped on the ALSPAC array with high quality imputation ($R^2$ score >0.9) on the GSA were selected. These variants were merged across the two arrays, converted to 'best-guess' genotypes and then included in a second joint imputation to the TOPMED reference panel. This method allowed high-quality imputation of both datasets. To address concerns about batch effects between cases and controls genotyped separately, 405 ALSPAC controls were also genotyped on the GSA alongside the HN5000 cases. This enabled sensitivity analyses to account for potential batch effects.

### Genetic ancestry stratification

Following the imputation process, markers from each imputation batch were filtered based on an imputation score of $R^2 > 0.8$ and merged across imputation batch and chromosomes. Markers were filtered for a call rate ≥ 0.98 and minor allele frequency (MAF) ≥1%. The major histocompatibility complex (MHC) region was removed, and the remaining markers were pruned for independent variants using linkage disequilibrium (LD) with a squared correlation ($r^2$) threshold of <0.2. This set of markers (N = 697,099) was utilised to compute kinship estimates between Individuals using the KING-robust kinship estimator[52] in PLINK 2.0[47]. The KING-robust method is specifically designed to be robust to population structure and admixture. It calculates kinship coefficients without being biased by the fact that certain populations may have different allele frequencies. In addition to the removal of 6679 known related individuals from the ALSPAC study, a kinship cutoff of >0.0884 was applied to exclude unexpected duplicates and individuals related at the second degree or closer. This cutoff is based on the geometric mean of the theoretical values for second and third-degree kinship, as outlined in the manual. Selection of related individuals or duplicates were prioritised based on either disease status (favouring cases over control) or array type (favouring newer arrays over older ones). After this process, 3441 individuals were excluded from the analysis. The remaining 58,625 individuals were classified into genetic ancestries using supervised ADMIXTURE analysis (ADMIXTURE 1.3[53]) with 75,164 common markers retained after quality control steps (Fig. S13). This assigns a percentage probability for belonging to each of the reference super-populations in the 1000 Genomes Project (N = 2504)[54]. We assigned individuals to a dominant genetic ancestry if their probability was ≥70% to any reference super-population. Of all individuals, 48,029 (83%) had a dominant genetic ancestry, while the remainder were classified as admixed. The distribution of individuals with a dominant genetic ancestry was as follows: 80.2% European (EUR), 0.1% Admixed Americans (AMR), 1.2% Africans (AFR), 1.3% South Asians (SAS), and 0.2% East Asians (EAS).

The remaining 17% were not able to be classified with a dominant genetic ancestry and were grouped as "admixed". To improve statistical power to detect risk loci across the relatively small sample sizes of non-European genetic ancestries, all five (AMR, AFR, SAS, EAS and admixed) were merged to create a "Mixed" group (N = 11,462) (Fig. S14a, b). Genome-Wide Association Studies (GWAS) were conducted separately in the European and Mixed ancestry samples and meta-analysed (see later). Principal Component Analysis (PCA) was carried out within each ancestral sample (European and Mixed) to assess population substructure and for covariate adjustment in GWAS (Fig. S15). For HLA fine-mapping analyses, a slightly different approach was required due to the region's high LD and highly correlated variants. Additionally, the HLA region is more susceptible to population substructure, making it challenging to identify causal variants that are consistent across ancestries. Therefore, for fine mapping, samples were grouped according to their dominant genetic ancestry (>70%) (EUR, AFR, and SAS) or admixed. Based on the homogeneous clustering identified through PCA (Fig. S16), the samples from Iran were separated into Middle Eastern (ME) ancestry. Small-sized numbers (Case/Control <50) of genetic ancestries (AMR and EAS) were merged into admixed. For each sample, PCA identified informative principal components (PCs) that showed significant associations ($p < 0.05$) with case-control status after adjusting for sex and imputation batch. These informative PCs, along with sex and imputation batch, were included as model covariates in the GWAS analysis.

## Association, meta- and conditional analysis

Using individual-level data, we conducted a mega-analysis within the ancestral groups and where possible subsequently meta-analysed across the two ancestry groups (Fig. S12).

Across the 9 arrays and 19 studies, there were several considerations in how to adjust for batch effects. Some studies, such as ARCAGE, were split across different arrays, such as the Oncoarray and AllofUs array. For other studies, such as UKBiobank, several arrays were used (UKBiLEVE and AffymetricUKB) (Supplementary Data 1). Finally, HN5000 and ALSPAC differed in their imputation as the 'double imputation' method was used. Therefore, a 'Batch' variable was created to represent the combination of studies, arrays and imputation approaches that could contribute to batch effects. To evaluate the potential impact of these different batches on the regression models, we conducted a sensitivity analysis by running GWAS within each batch and assessed heterogeneity using METAL[55]. We excluded markers with a heterogeneity *p*-value $<5 \times 10^{-8}$, resulting in the removal of 137 markers in the European sample GWAS. We compared the allele frequencies within the case and controls across each batch to confirm their comparability (Supplementary Data 15).

Association analysis was conducted separately for all sites combined and for each HNSCC subsite using PLINK. Analyses were logistic regressions for each variant, adjusted for sex, batch and informative PCs, which were selected *based* on their significant associations ($p < 0.05$) with case-control status after adjusting for sex and imputation batch. Association tests were first run within the European and Mixed groups and then meta-analysed with METAL[55] using a fixed effects model to identify cross-ancestral loci (Fig. S12). There was minimal inflation after adjustment for informative PCs in most analyses (λ ranging from 1.00 to 1.03). However, the HPV(+) and HPV(−) OPC analyses for the Mixed group did show evidence of inflation (HPV(+) OPC: λ = 1.12; HPV(·) OPC: λ = 1.20) (Fig. S17). Consequently, rather than a meta-analysis, the GWAS analysis for OPC was conducted only in the European sample, with consistency of top SNPs assessed separately in the Mixed sample. For all other subsites, loci that achieved *p*-value $<5 \times 10^{-8}$ in the meta-analysis were referred to as cross-ancestral. This threshold was selected as it is equivalent to a standard Bonferroni correction for one million independent tests. Loci satisfying $p < 5 \times 10^{-8}$ within each

ancestral sample, which 1) were not significant in the meta-analysis and 2) showed no attenuation upon conditional analysis of nearby lead cross-ancestral SNPs and therefore considered to be independent from the cross-ancestral SNP, were hereby referred to as ancestral-specific (Supplementary Data 16). Where these existed in the Mixed ancestry sample, further stratification into the five dominant genetic ancestries was performed. Regional association plots were generated using Locus Zoom (https://my.locuszoom.org/).

We utilised METASOFT[56], a meta-analysis software, to generate Posterior Mean (PM) plots for visualising the association between genetic variants and the effect sizes at specific subsites. To assess whether an effect is consistent across subsites, we obtained the posterior probability (m-value) from METASOFT. PM-plots were then generated by plotting the m-value against the $-\log_{10}$ p-value, which was derived from the subsite-specific meta-analysis. This visualisation provides an intuitive way to evaluate both the consistency of effects and their statistical significance across subsites.

## HLA fine mapping

Variants that were directly genotyped in chromosome 6 were extracted from genotyping data of all arrays and standardised to hg19 using LiftOver. Due to restrictions in data access from ALSPAC, additional data from the UK Biobank were used to replace ALSPAC for double imputation with HN5000 as described above. Per variant QC was conducted by deduplication of SNP data, strand alignment, removal of palindromic variants (i.e., SNPs with A/T or G/C alleles), removal of poor-quality variants with a missingness threshold of 10% and a Hardy-Weinberg equilibrium threshold of $1 \times 10^{-10}$. Sample QC was conducted after the removal of samples with high missingness rates, outlier heterozygosity, discordant sex information, and genetically identical samples. A flow diagram of QC steps for the HLA fine mapping is provided in Figure S18.

The HLA region (Chromosome 6:28Mb–34Mb) was imputed for SNPs and classical HLA class I and II alleles using the Michigan imputation server with the most recent HLA Multi-ethnic reference panel (Four-digit Multi-ethnic HLA v2)[57]. Only high-quality SNPs, alleles or amino-acid residues were included in the analysis (imputation $r^2 > 0.95$). The final set of imputed variants used in association analyses was of high quality; 91% of the variants and 71% of the less common variants (MAF <0.05) had imputation $R^2 \geq 0.95$. HLA-wide association analysis was conducted, controlling for sex, informative PCs, and imputation batch (described above), and meta-analysed with a random effect model using PLINK[47] to identify cross-ancestral variants. Any genetic ancestries with fewer than 50 samples were excluded from meta-analyses due to power. Stepwise conditional analysis was conducted to identify independent variants within each ancestry, where variants with the lowest p-value after each round were added to the subsequent model, and the analysis was repeated until no further variants met the significance threshold. As HLA fine mapping was conducted independently from GWAS, a probability threshold was set to $2.4 \times 10^{-6}$. This was based on the total number of imputed HLA variants (0.05/20,762), which included SNPs, amino acid variants, and classical HLA alleles after quality control as described previously[58].

To identify haplotypes associated with risk within each subsite that were linked to the top novel variants identified from fine mapping, the haplo.stats package v.1.9.5.1 in R was applied to identify combinations of HLA 4-digit alleles within each population. The haplo.em and haplo.glm algorithms identified haplotype candidates in each population with a minimum haplotype frequency threshold set at 0.01 in comparison to the most common haplotype within the ancestry. Haplotype candidates that were in high LD ($r^2 > 0.8$) with variants from fine mapping were then tested for association with risk using the meta-analysis approach to determine if they conferred similar risk compared to their variant counterparts.

## Testing for Independence and functional equivalents of lead variants

Variants identified in each HNSCC subsite analysis from the GWAS and fine mapping were compared across subsites to evaluate whether they were linked or independent. This was also performed to define variants that were novel compared to previously reported signals and to determine overlapping signals between cross-ancestral and population-specific variants. LD was measured by $r^2$ using PLINK 1.9[47] within the overall dataset. If LD > 0.3, then conditional analysis was performed to evaluate if the significance of the variant of interest attenuated to lower than $2.4 \times 10^{-6}$. If both criteria were met, variants were considered to be dependent.

To determine functional equivalents of the variants identified through fine mapping, amino acid changes, alleles and haplotypes that were in moderate to high LD ($r^2 > 0.5$) with lead novel variants were further evaluated. Effect sizes and significance levels were compared when replacing the lead variant with the related variant in the fully adjusted cross-ancestral model. Bayesian Information Criterion (BIC) were then evaluated to compare the model fit of the original model with the lead variants identified from fine mapping to the model with the related variant replacing the original lead variant. Every permutation of variants was considered to determine if one variant could replace by another and still provide the same information as the original lead variant.

## Stratified analyses

For each independent top hit identified in GWAS and HLA fine mapping, the analysis was repeated, stratified by sex, smoking status, drinking status, geographic region, and within all cancer subsites separately. The effects across strata were assessed for heterogeneity using the $\chi^2$-based Q test (Cochran's Q test) using R (v4.1.2). Further stratification related to smoking and alcohol was conducted in non-HPV-related cancers. This assessed effects in never-smoking non-drinkers, smoking non-drinkers, never-smoking drinkers and ever-smoking drinkers to assess the independence of these risk factors where data was available. Results were presented in forest plots (Figs. S7, S9).

## Heritability and genetic correlation

SNP-based heritability was estimated in the European and Mixed ancestry samples using linkage disequilibrium score regression (LDSR)[59] using the Complex-Traits Genetics Virtual Lab platform[60]. To assess the contribution of HLA variants to HNSCC heritability and its subsites, we quantified the proportion of variance in cancer risk explained by the sentinel variants in the HLA region identified from the GWAS, thereby evaluating their relative contribution to the total regional variance. Heritability estimates in the Mixed ancestry sample are not presented in the main manuscript due to the heterogeneous nature of these samples, which makes estimates of heritability unreliable. These are provided in Supplementary Data 6 for completeness.

## Colocalization of GWAS-identified variants with molecular QTLs and lifestyle risk factors

Colocalization of genetic associations between all identified top hit variants from GWAS analyses outside of HLA regions and their gene expression and related traits was calculated using default LDs and a window size of ±75 kb using the COLOC package[61]. All colocalization analyses were conducted using HNSCC data of European ancestry. Expression quantitative trait loci (eQTLs) in whole blood were obtained from the eQTLGen Consortium[62] due to its role in immune response and systemic inflammation. eQTLs in oesophagus and lung tissues, as well as splicing QTLs (sQTLs) in oesophagus and lung tissues and whole blood, were sourced from the Genotype-Tissue Expression (GTEx) project (v8)[63], given their anatomical proximity and shared risk factors,

such as tobacco and alcohol exposure[64]. Additionally, DNA methylation QTL (mQTL) data from whole blood and lung tissue were sourced from GTEx[64,65]. Colocalization analysis was performed at genetic loci associated with HNSCC risk. Loci were considered eligible for assessment with colocalization if they harboured at least one variant significantly associated with expression or splicing (eQTL/sQTL; FDR-adjusted $p < 0.05$) or with DNA methylation levels (mQTL; $p < 5 \times 10^{-8}$). For each eligible locus, colocalization analysis was then performed using all SNPs available within the locus. Summary statistics from GWAS for smoking and alcohol consumption behaviours were sourced from the GWAS & Sequencing Consortium on Alcohol and Nicotine Use (GSCAN) 67. The analysis considers the posterior probability of colocalization for a single shared variant responsible for the associations in both traits (posterior probability for hypothesis 4 (PP4)). Values over 0.7 were considered strong evidence of colocalization. Where the lead variant was not available in the LD reference panel required for COLOC, the variant with the highest LD was used instead.

## Technical validation

For the technical validation of the imputed *TP53* variant, we utilised a Taqman assay to genotype this specific variant in a subset of samples from the Central and Eastern European Study (CEE) and ARCAGE studies. Individuals removed from the GWAS in QC steps or those with technical issues during the Taqman assays, e.g., failure to amplify, were removed, resulting in 2370 samples where consistency could be assessed. Overall concordance and non-reference discordance were calculated.

## Reporting summary

Further information on research design is available in the Nature Portfolio Reporting Summary linked to this article.

## Data availability

The full GWAS summary statistics have been deposited in the MRC IEU OpenGWAS database and will be publicly available at https://gwas.mrcieu.ac.uk/ under accession numbers ieu-b-5129 [https://opengwas.io/datasets/ieu-b-5129] (head and neck cancer), ieu-b-5130 [https://opengwas.io/datasets/ieu-b-5130] (hypopharyngeal cancer), ieu-b-5131 [https://opengwas.io/datasets/ieu-b-5131] (laryngeal cancer), ieu-b-5132 [https://opengwas.io/datasets/ieu-b-5132] (oral cavity cancer), ieu-b-5133 [https://opengwas.io/datasets/ieu-b-5133] (HPV-negative oropharyngeal cancer) and ieu-b-5134 [https://opengwas.io/datasets/ieu-b-5134] (HPV-positive oropharyngeal cancer). The individual-level genotype data analysed in this study are available through dbGaP under restricted access due to participant privacy and informed consent limitations. Access can be obtained by applying through the dbGaP portal, subject to approval by the relevant Data Access Committees. The following datasets were used: phs001273.v4.p2 [https://www.ncbi.nlm.nih.gov/projects/gap/cgi-bin/study.cgi?study_id=phs001273.v4.p2] (OncoArray Consortium – Lung Cancer Studies) phs001202.v2.p1 [https://www.ncbi.nlm.nih.gov/projects/gap/cgi-bin/study.cgi?study_id=phs001202.v2.p1] (OncoArray: Oral and Pharynx Cancer) phs001173.v1.p1 [https://www.ncbi.nlm.nih.gov/projects/gap/cgi-bin/study.cgi?study_id=phs001173.v1.p1] (NCI Head and Neck Cancer Study, HumanOmniExpress-12v1.0) phs002503.v1.p1 [https://www.ncbi.nlm.nih.gov/projects/gap/cgi-bin/study.cgi?study_id=phs002503.v1.p1] (GWAS of Oral Cavity, Pharynx, and Larynx Cancers in European, North, and South American populations) Access to each dbGaP dataset requires submission of a Data Access Request through https://dbgap.ncbi.nlm.nih.gov/aa/wga.cgi?page=login. Approval is granted by the respective Data Access Committees, and responses are typically issued within 2–4 weeks. Data are available for use in biomedical research consistent with the consent of the study participants. Data from Head and Neck 5000, UK Biobank and ALSPAC cohorts are available under restricted access through their

respective data access procedures: Head and Neck 5000, UK Biobank [https://www.ukbiobank.ac.uk/enable-your-research/apply-for-access] ALSPAC. These data are subject to ethical and legal restrictions. Requests must be submitted directly to the data custodians through the links above. Response times and access terms are determined by the respective institutions. Source data are provided with this paper.

## Code availability

This study did not employ any custom code. Instead, it utilised publicly available software tools for genetic analyses, which are cited throughout the manuscript and reporting summary.

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

## Acknowledgements

This study was funded in part by the European Union's Horizon 2020 research and innovation programme under grant agreement No 825771 (PB, SV) (HEADSpAcE project) and by the US National Institute of Dental and Craniofacial Research (NIDCR) grants R03DE030257 (SV) and R01DE025712 (PB, BD). Genotyping using the Oncoarray and the All of Us array was performed at Centre for Inherited Disease (CIDR) and funded by NIDCR 1X01HG007780-0 (PB) and jointly by NIDCR/NCI X01HG010743 (SV). This publication presents data from the Head and Neck 5000 study. The study was a component of independent research funded by the National Institute for Health and Care Research (NIHR) under its Programme Grants for Applied Research scheme (RP-PG-0707-10034). The views expressed in this publication are those of the author(s) and not necessarily those of the NHS, the NIHR or the Department of Health. Core funding was also provided through awards from Above and Beyond, University Hospitals Bristol and Weston Research Capability Funding and the NIHR Senior Investigator award to Professor Andy Ness. Round 1 genotyping was funded by a US National Institute of Dental and Craniofacial Research (NIDCR) grant 1X01HG007780-0. Round 2 genotyping was funded by World Cancer Research Fund Pilot Grant (grant number: 2018/1792), Above and Beyond Charity (GA2500), Wellcome Trust Research Training Fellowship (201237/Z/16/Z) and Cancer Research UK Programme Grant, the Integrative Cancer Epidemiology Programme (grant number: C18281/A19169). This latter grant also supported Human papillomavirus (HPV) serology. This research has been conducted using the UK Biobank Resource under Application Number 40644. The work of Tom Dudding was supported by an NIHR Clinical Lectureship (CL-2022-25-007). The work of Dr. Polesel is partially supported by the Italian Ministry of Health 'Ricerca Corrente'. The University of Pittsburgh head and neck cancer case-control study is supported by US National Institutes of Health grants P50CA097190, P30CA047904 and R01DE025712 (PB, BD). Geoffrey Liu is the M. Qasim Choksi Research Chair in Translational Research at University Health Network and University of Toronto and is supported by the Princess Margaret Head and Neck Translational Programme, which is supported by philanthropic funds from the Wharton Family, Joe's Team, Gordon Tozer, Reed Fund, and the Riley Family. The University of North Carolina studies were supported in part by grants CA61188 (AO) and CA90731 (AO) from the National Institutes of Health. Northern Cancer Foundation (Principal Investigator Grants to MSC Conlon, DP Saunders). Rayjean J. Hung is the CIHR Canada Research Chair, and the study is supported by the Canadian Cancer Society and the Canadian Institute of Health Research. Tim Waterboer serves on advisory boards for MSD (Merck) Sharp & Dohme. Scott V Bratman is an inventor on patents related to cell-free DNA mutation and methylation analysis technologies that are unrelated to this work and have been licensed to Roche and Adela, respectively. Scott V Bratman is a co-founder of and has ownership in Adela. The authors would like to thank all the patients and their families involved in these studies. Where members are identified as personnel of the International Agency for Research on Cancer/ World Health Organisation, the authors alone are responsible for the views expressed in this article, and they do not necessarily represent the decisions, policy or views of the International Agency for Research on Cancer / World Health Organisation.

## Author contributions

S.V. and T.D. conceived and led the project, providing overall direction, coordination, and supervision throughout all stages. E.E. and A.S. led the data analysis and manuscript writing, with E.E. leading the work as first author. H.A.P. and J.M. contributed substantially to data analysis and interpretation and were closely involved in manuscript preparation. V.G. supported data analysis. N.T., P.B., A.F.I., M.G., B.D., and S.H. advised on study design, data interpretation, and contextualisation. W.A., L.Ale., L.M.R.B.A., J.B., S.V.B., C.C., M.S.C.C., D.I.C., M.C., M.Cur., A.d.C., Jd.O., M.H., C.M.H., I.H., R.J.H., L.P.K., P.Lag., A.Lag., G.L., G.J.M., A.F.O., S.P., L.F.P., J.V.P., J.P., M.P., H.R., R.R.G., L.R., M.R., P.A.R.U., S.A.S., D.P.S., S.C.S.L., M.V., S.V.Z., T.W., K.Z., and A.Z. contributed primary data and biospecimens. All authors reviewed and approved the final manuscript.

## Competing interests

All Authors declare no competing interests.

## Additional information

Elmira Ebrahimi [1,2], Apiwat Sangphukieo[1,3], Hanla A. Park [1], Valerie Gaborieau[1], Aida Ferreiro-Iglesias[1], Brenda Diergaarde[4,5], Wolfgang Ahrens [6], Laia Alemany[7,8,9], Lidia MRB Arantes[10], Jaroslav Betka[11], Scott V. Bratman[12], Cristina Canova[13], Michael SC Conlon[14], David I. Conway[15], Mauricio Cuello[16], Maria Paula Curado [17], Ana Carolina de Carvalho[1], Jose Carlos de Oliviera[18], Mark Gormley [19], Maryam Hadji[2,20], Sarah Hargreaves [21], Claire M. Healy [22], Ivana Holcatova [23], Rayjean J. Hung [24,25], Luis P. Kowalski[26,27], Pagona Lagiou[28], Areti Lagiou[29], Geoffrey Liu[30], Gary J. Macfarlane[31], Andrew F. Olshan[32], Sandra Perdomo [1], Luis Felipe Ribiero Pinto[33], Jose Roberto V. Podesta [34], Jerry Polesel [35], Miranda Pring[19], Hamideh Rashidian[2], Ricardo R. Gama [36], Lorenzo Richiardi[37], Max Robinson [38], Paula A. Rodriguez-Urrego [39], Stacey A. Santi[40], Deborah P. Saunders[41], Sheila C. Soares-Lima [42], Nicholas Timpson [43], Marta Vilensky[44], Sandra V. von Zeidler [45], Tim Waterboer [46], Kazem Zendehdel[2], Ariana Znaor [47], Paul Brennan[1], HEADSpAcE Consortium*, James McKay[1], Shama Virani [1,105] ✉ & Tom Dudding [19,105] ✉

[1]Genomic Epidemiology Branch, International Agency for Research on Cancer (IARC/WHO), Lyon, France. [2]Cancer Research Center, Cancer Institute, Tehran University of Medical Sciences, Tehran, Iran. [3]Center of Multidisciplinary Technology for Advanced Medicine (CMUTEAM), Faculty of Medicine, Chiang Mai University, Chiang Mai, Thailand. [4]Department of Human Genetics, School of Public Health, University of Pittsburgh, Pittsburgh, USA. [5]UPMC Hillman Cancer Center, Pittsburgh, USA. [6]Leibniz Institute for Prevention Research and Epidemiology-BIPS, Bremen, Germany. [7]Catalan Institute of Oncology. ICO, L'Hospitalet, Barcelona, Spain. [8]Bellvitge Biomedical Research Institute (IDIBELL), L'Hospitalet, Barcelona, Spain. [9]CIBER en Epidemiología y Salud Pública (CIBERESP), Madrid, Spain. [10]Barretos Cancer Hospital, Barretos, Brazil. [11]Department of Otorhinolaryngology and Head And Neck Surgery, 1.st Medical Faculty, Charles University, Faculty Hospital Motol, Prague, Czech Republic. [12]Depts of Radiation Oncology, Princess Margaret Cancer Centre, University of Toronto, Toronto, Canada. [13]Unit of Biostatistics, Epidemiology and Public Health, Department of Cardio-Thoraco-Vascular Sciences and Public Health, University of Padova, Padova, Italy. [14]Epidemiology, Outcomes & Evaluation Research, Health Sciences North Research Institute, Sudbury, Canada. [15]School of Medicine, Dentistry and Nursing, University of Glasgow, Glasgow, UK. [16]Oncology, Hospital de Clinicas Dr. Manuel Quintela, Montevideo, Uruguay. [17]Epidemiology and Statistics Group, Research Center, A.C Camargo Cancer Center, São Paulo, Brazil. [18]Araújo Jorge Cancer Hospital, Associação de Combate ao Câncer em Goiás, Goiania, Brazil. [19]Bristol Dental School, Bristol University, Bristol, UK. [20]A.I. Virtanen Institute for Molecular Sciences, University of Eastern Finland, Kuopio, Finland. [21]University Hospitals Bristol and Weston NHS Foundation Trust, Bristol, UK. [22]School of Dental Science, Dublin Dental University Hospital, Trinity College Dublin, Dublin, Ireland. [23]Institute of Hygiene & Epidemiology, 1st Faculty of Medicine, Charles University, Prague, Czech Republic. [24]Prosserman Centre for Population Health Research, Lunenfeld-Tanenbaum Research Institute, Sinai Health System, Toronto, Canada. [25]Division of Epidemiology, Dalla Lana School of Public Health, University of Toronto, Toronto, Canada. [26]Department of Head and Neck Surgery, University of São Paulo Medical School, São Paulo, Brazil. [27]Department of Head and Neck Surgery and Otorhinolaryngology, A C Camargo Cancer Center, São Paulo, Brazil. [28]Department of Hygiene, Epidemiology & Medical Statistics, School of Medicine, National and Kapodistrian University of Athens, Athens, Greece. [29]Department of Public and Community Health, School of Public Health, University of West Attica, Athens, Greece. [30]Medicine, Epidemiology, Medical Oncology, Princess Margaret Cancer Centre, University of Toronto, Toronto, Canada. [31]Epidemiology Group, School of Medicine, Medical Sciences and Nutrition, University of Aberdeen,

Aberdeen, UK. [32]Department of Epidemiology, Gillings School of Global Public Health, University of North Carolina, Chapel Hill, USA. [33]Programa de Carcinogênese Molecular, Instituto Nacional de Câncer - INCA, Rio de Janeiro, Brazil. [34]Head and Neck Surgery Division, Women's Association for Education and Fight Against Cancer/AFECC, Vitória, Brazil. [35]Unit of Cancer Epidemiology, Centro di Riferimento Oncologico di Aviano (CRO) IRCCS, Aviano, Italy. [36]Department of Head and Neck Surgery, Barretos Cancer Hospital, São Paulo, Brazil. [37]Cancer Epidemiology Unit, University of Turin, Turin, Italy. [38]Cellular Pathology, The Newcastle upon Tyne Hospitals NHS Foundation Trust, Newcastle upon Tyne, UK. [39]Pathology and Laboratories, Pathology, University Hospital Fundacion Santa Fe de Bogota, Bogota, Colombia. [40]Clinical Oncology Research, Health Sciences North Research Institute, Sudbury, Canada. [41]Department of Dental Oncology, Health Sciences North, Northern Ontario School of Medicine University, Sudbury, Canada. [42]Brazilian National Cancer Institute, Rio de Janeiro, Brazil. [43]MRC Integrative Epidemiology Unit, Bristol University, Bristol, UK. [44]Instituto de Oncologia Angel H Roffo, Universidad de Buenos Aires, Buenos Aires, Argentina. [45]Pathology Department, Federal University of Espírito Santo, Vitória, Brazil. [46]Infections and Cancer Epidemiology, German Cancer Research Center (Deutsches Krebsforschungszentrum, DKFZ), Heidelberg, Germany. [47]Cancer Surveillance Branch, International Agency for Research on Cancer (IARC/WHO), Lyon, France. [105]These authors contributed equally: Shama Virani, Tom Dudding.
✉e-mail: viranis@iarc.who.int; tom.dudding@bristol.ac.uk

## HEADSpAcE Consortium

Elmira Ebrahimi [1,2], Apiwat Sangphukieo [1,3], Hanla A. Park [1], Valerie Gaborieau[1], Aida Ferreiro-Iglesias[1], Brenda Diergaarde[4,5], Wolfgang Ahrens [6], Laia Alemany[7,8,9], Lidia MRB Arantes[10], Jaroslav Betka[11], Scott V. Bratman[12], Cristina Canova[13], Michael SC Conlon[14], David I. Conway[15], Mauricio Cuello[16], Maria Paula Curado [17], Ana Carolina de Carvalho[1], Jose Carlos de Oliveira[18], Mark Gormley[19], Maryam Hadji[2,20], Sarah Hargreaves [21], Claire M. Healy [22], Ivana Holcatova [23], Rayjean J. Hung [24,25], Luis P. Kowalski[26,27], Pagona Lagiou[28], Areti Lagiou[29], Geoffrey Liu[30], Gary J. Macfarlane[31], Andrew F. Olshan[32], Sandra Perdomo [1], Luis F. Pinto[33], Jose Roberto V. Podesta [34], Jerry Polesel [35], Miranda Pring[19], Hamideh Rashidian[2], Ricardo R. Gama [36], Lorenzo Richiardi [37], Max Robinson [38], Paula A. Rodriguez-Urrego [39], Stacey A. Santi[40], Deborah P. Saunders[41], Sheila C. Soares-Lima [42], Nic Timpson[43], Marta Vilensky[44], Sandra V. von Zeidler[45], Tim Waterboer [46], Kazem Zendehdel[2], Ariana Znaor [47], Paul Brennan[1], James McKay[1], Shama Virani [1,105]✉, Tom Dudding [19,105]✉, Roque Adam[48], Antonio Agudo[49], Salima Alibhai[50], Shaymaa F. AlWaheidi[1], Miquel Angel Pavon[49], Namrah Anwar[51], Paola Engelmann Arantes[52], Lisa Arguello[53], Yubelly Avello[54], Lucas Avondet[48], Ana M. Baldión-Elorza[54], Camila Batista Daniel[55], Bianca Beraldi[56], Barbara Berenstein[57], Patricia Bernal[58], Natália Bernardino Rodrigues[59], Josipa Bilic Zimmermann[49], Marianna G. Botta[52], Lourine Bouvard[1], Jesús Brenes[49], Nicole Brenner[60], Carol Brentisci[61], Catalina Burtica[54], María L. Cabañas[62], Erick Cantor[63], Raiany S. Carvalho[64], Andre L. Carvalho[65], Luigi Chiusa[66], Priscilla Chopard[1], Qurratulain Chundriger[67], Omar Clavero[49], Isabela Costa[68], Grant Creaney[15], Cecilia Cuffini[69], Tauana C. Dias[64], Evandro Duccini de Souza[56], Lais C. Durant[70], Alberto Escallón[71], Gisele Aparecida Fernandes[52], Béatrice Fervers[72], Valentina Fiano[73], Frederico Firme Figueira[59], Regina Furbino Villefort[59], Manuela Gangemi[61], Paolo Garzino-Demo[74], Mahin Gholipour[75], Raul Giglio[57], Mariel A. Goulart[15], Jéssica Graça Sant'Anna[55], Marek Grega[76], Anna Clara Gregório Có[55], Arnau Guasch[49], Jose A. Hakim[71], David N. Hayes[77], Marco Homero de Sá Santos[56], Katrina Hurley[78], Magalí Insfran[79], Giuseppe C. Iorio[80], Moghira Iqbaluddin Siddiqui[81], Jannik Johannsen[82], Martin Kaňa[83], Jens Peter Klussmann[82], Evelio Legal[84], Jeferson Lenzi[56], Fernando Luiz Dias[68], Iván Lyra González[85], Willene Machado Zorzaneli[55], Ricardo Mai Rocha[56], Manel Mañós[49], Priscila Marinho de Abreu[55], Maryam Marzban[86,87], James McCaul[88], Alex D. McMahon[15], Carlos Mena[84], Elismauro F. Mendonça[89], Laura Mendoza[79], Lorena Meza[79], Birgitta Michels[60], Matinair S. Mineiro[89], Chiara Moccia[73], Pamela Mongelos[79], Ana L. Montealegre-Páez[90], Francisca Morey Cortes[49], Alvaro Muñoz[91], Andy Ness[78], Aline B. Neves[52], Marco Oliva[49], José Carlos de Oliveira[92], Hernán Ortiz[53], José Ortiz[84], Marta Osorio[84], Vanessa Ospina[63], Oliviero Ostellino[93], Mauricio Palau[54], Claire Paterson[94], Sonia Paytubi Casabona[49], Giancarlo Pecorari[74], David M. Pereira[95], Olivia Pérol[72], Shahid Pervez[67], Alicia Pomata[62], Maja Popovic[73], Alisson Poveda[90], Carol P. Prado[52], Kristina M. Prager[60], Guglielmo Ramieri[74], Saida Rasul[50], Juliana NI Rego[96], Rui M. Reis[64], Helene Renard[1], Umberto Ricardi[80], Giuseppe Riva[74], Frederic Rodilla[49], Ingrid Rodriguez[97], María I. Rodríguez[79], Alastair Ross[15], Pierre-Eric Roux[98], Tazeen Saeed Ali[99], Pierre Saintigny[100], Juan J. Santivañez[71], Cristóvam Scapultampo-Neto[101], Javier Segovia[63], Agenor Sena[56], Ricardo Serrano[84], Shachi J. Sharma[82], Oliver Siefer[82], Stephanie Smart[102], Bruna P. Sorroche[64], Cinthia Sosa[62], Juliana Souza de Oliveira[17], Antonella Stura[61], Steven Thomas[78], Oscar Torres[103], Sara Tous[49], Gonzálo Ucross[58], Adriana Valenzuela[79], José Roberto Vasconcelos de Podestá[56], Alex Whitmarsh[78] & Sylvia Wright[104]

[48]H&N cancer Department, Universidad de Buenos Aires, Ciudad Autonoma de Buenos Aires, Argentina. [49]Catalan Institute of Oncology (ICO), Barcelona, Spain. [50]Department of Surgery, Dental Hygiene Program, Aga Khan University Hospital, Karachi, Pakistan. [51]Faculty of Science and Technology, University of Central Punjab, Lahore, Pakistan. [52]Group of Epidemiology and Statistics on Cancer, A.C. Camargo Cancer Center, Sao Paolo, Brazil. [53]Servicio de Cabeza y Cuello, Instituto Nacional del Cáncer, Ministerio de Salud Pública y Bienestar Social, Capiatá, Paraguay. [54]Pathology and Laboratory Department, Fundación SantaFe de Bogotá, Bogotá, Colombia. [55]Postgraduate Program in Biotechnology, Universidade Federal do Espirito Santo, Vitoria, Brazil. [56]Head and Neck Surgery Division, Associação Feminina de Educação e Combateao Câncer(AFECC), Hospital Santa Rita de Cássia, Vitoria, Brazil. [57]H&N cancer

Department, Institute of Oncology Angel H. Roffo, University of Buenos Aires, Ciudad Autonoma de Buenos Aires, Argentina. [58]Department of Radiology, Division of Nuclear Medicine, Fundación SantaFe de Bogotá, Bogotá, Colombia. [59]Department of Pathology, Universidade Federal do Espirito Santo, Vitoria, Brazil. [60]Division of Infections and Cancer Epidemiology, German Cancer Research Center (DKFZ), Heidelberg, Germany. [61]Department of Medical Sciences, Cancer Epidemiology Unit, AOU Città della Salute e della Scienza di Torino, Turin, Italy. [62]Departamento de Anatomía Patológica, Instituto Nacional del Cáncer, Ministerio de Salud Pública y Bienestar Social, Capiatá, Paraguay. [63]Oncology Department, Fundación SantaFe de Bogotá, Bogotá, Colombia. [64]Molecular Oncology Research Center, Barretos Cancer Hospital, Barretos, Brazil. [65]Department of Head and Neck Surgery, Barretos Cancer Hospital, Barretos, Brazil. [66]Pathology Unit, AOU Città della Salute e della Scienza di Torino, Turin, Italy. [67]Department of Pathology and Laboratory Medicine, Section of Histopathology, Aga Khan University Hospital, Karachi, Pakistan. [68]INCA, Rio de Janeiro, Brazil. [69]Universidad Nacional de Cordoba, Cordoba, Argentina. [70]A.C Camargo Cancer Center, São Paulo, Brazil. [71]Department of Surgery, Head and Neck Division, Fundación SantaFe de Bogotá, Bogotá, Colombia. [72]Department Cancer Environnement, Centre Léon Bérard, Lyon, France. [73]Department of Medical Sciences, Cancer Epidemiology Unit, University of Turin, Turin, Italy. [74]Department of Surgical Sciences, University of Turin, Turin, Italy. [75]Metabolic Disorders Research Center, Golestan University of Medical Sciences, Gorgan, Iran. [76]Department of Pathology and Molecular Medicine, 2nd Faculty of Medicine, Charles University and Motol University Hospital, University Hospital in Motol, Prague, Czech Republic. [77]Department of Genetics, Genomics and Informatics, University of Tennessee Health Science Center, Memphis, USA. [78]Bristol Dental School, University of Bristol, Bristol, United Kingdom. [79]Salud Pública, Instituto de Investigaiones en Ciencias de la Salud (IICS), Universidad Nacional de Asunción (UNA), San Lorenzo, Paraguay. [80]Department of Oncology, University of Turin, Turin, Italy. [81]Department of Surgery, Section of E.N.T, Aga Khan University Hospital, Karachi, Pakistan. [82]Department of Otorhinolaryngology, Head and Neck Surgery, University of Cologne, Cologne, Germany. [83]Departrment of Otorhinolaryngology and Head and Neck Surgery, University Hospital in Motol, Prague, Czech Republic. [84]Cátedra Otorrinonaringología, Hospital de Clínicas, Facultad de Ciencias Médicas, Universidad Nacional de Asunción, San Lorenzo, Paraguay. [85]Servicio de Oncología Clínica Hospital de Clínicas, Universidad de la República, Montevideo, Uruguay. [86]The Persian Gulf Tropical Medicine Research Center, The Persian Gulf Biomedical Sciences Research Institute, Bushehr University of Medical Sciences, Bushehr, Iran. [87]Statistics Genetic Lab, QIMR, Berghofer Medical Research Institute, Brisbane, Australia. [88]Department of Oral and Maxillofacial/Head and Neck Surgery, NHS Greater Glasgow and Clyde, Glasgow, United Kingdom. [89]Hospital Câncer Araújo Jorge, Goiânia, Brazil. [90]Faculty of Medicine, El Bosque University, Bogotá, Colombia. [91]Oncology Department, Division of Radiotherapy, Fundación SantaFe de Bogotá, Bogotá, Colombia. [92]Goiânia Cancer Registry (BR), Goiânia, Brazil. [93]Department of Oncology, Division of Medical Oncology, AOU Città della Salute e della Scienza di Torino, Turin, Italy. [94]Beatson West of Scotland Cancer Centre, NHS Greater Glasgow and Clyde, Glasgow, United Kingdom. [95]Radiation Oncology Department, Institute of Oncology Angel H. Roffo, University of Buenos Aires, Ciudad Autonoma de Buenos Aires, Argentina. [96]Clinical Research Center, Associação Feminina de Educação e Combateao Câncer(AFECC), Hospital Santa Rita de Cássia, Vitoria, Brazil. [97]Laboratorio de Anatomía Patológica, Hospital de Clínicas, Facultad de Ciencias Médicas, Universidad Nacional de Asunción, San Lorenzo, Paraguay. [98]Department of Surgery, Centre Léon Bérard, Lyon, France. [99]School of Nursing and Midwifery, Aga Khan University Hospital, Karachi, Pakistan. [100]Centre Léon Bérard, Lyon, France. [101]Pathology and Molecular Diagnostics Service, Barretos Cancer Hospital, Barretos, Brazil. [102]NHS Greater Glasgow & Clyde, Glasgow, United Kingdom. [103]Radiology Department, Fundación SantaFe de Bogotá, Bogotá, Colombia. [104]Institute of Cancer Sciences, University of Glasgow, Glasgow, United Kingdom.

