## [Peer Review File · Nature Communications]

Cross-ancestral GWAS identifies 29 variants across head and neck cancer subsites

Corresponding Author: Dr Tom Dudding

Version 0:

Reviewer comments:

Reviewer #1

(Remarks to the Author)

The authors performed a multi-ancestry GWAS and refined the genetic map of head and neck cancer as well as its subsites. Novel variants were found, additionally advancing the understanding of genetic factors on HNSCC development. I have some concerns for the study design and methods for this study.

1. Study Design and Population:

(1) This study includes samples genotyped using 9 different arrays across 19 studies, with some studies containing only case or control samples (e.g., the Sudbury study and the HN5000 study). This creates a challenge in constructing a batch effect variable that accounts for both array and study effects while preserving the distinction between cases and controls. The minor allele frequencies of the identified SNPs in cases and controls from different studies should be further examined to ensure that the observed associations are not influenced by batch effects.

(2) The authors used UK biobank samples but included only 14435 controls. How were these controls selected?

2. Analysis:

(1) The author performed cross-ancestry meta-analysis by using a fixed effect model in METAL. I think a random effect model like MR-MEGA should be used to consider the different standard errors of SNPs in different ancestries.

(2) Statistical Analysis: Pleiotropic effect and site-specific effect of SNPs could be explored, for example, by ASSET and metasoftware, and be better presented.

(3) The results of the previously identified HNSCC risk variants in this study should be presented.

(4) In addition to eQTL data, other functional annotation should be performed. For example, whether the identified SNPs are located in the chromatin/transcription active region, how they regulate the target SNPs.

(5) The interaction between SNPs and smoking/drinking should be further investigated. Specifically, it would be valuable to explore whether the effects of smoking and drinking differ among individuals with varying genetic risk profiles. Such an analysis could offer valuable insights into personalized prevention strategies for HNSCC.

(6) Given the complexity of HLA region, conditional analysis on the previously reported HLA variants should be performed to confirm the "novel" role of the newly identified HLA variants.

(7) Are the allele frequencies and the LD structure of the identified HLA variants, particularly those ancestry specific, different among different ancestral groups?

(8) How much is the contribution of rare and common variants, HLA and non-HLA variants, the reported variants and the novel variants, to the heritability of HNSCC and different subtypes?

(9) The EAF of some of the 18 novel variants are not shown in Table 1. Please provide the detailed information.

(10) Experimental validation should be performed to confirm the effect of SNP on gene expression, for example, rs78378222 on TP53 expression and its effect on the impairment of miRNA binding, rs28419191 and rs1131769 on CTNNA1 expression.

3. Explanation and others:

(1) How do the authors explain that the heritability of HPC+ oropharyngeal cancer is significantly higher than that of HPC- oropharyngeal cancer? It appears that virological factors may contribute more to the development of HPC+ oropharyngeal cancer.

(2) Biological explanation should be included for the site specific or HPV+/HPV- specific SNPs.

(3) The results section of the manuscript should be organized in a clearer and more concise manner.

Reviewer #2

(Remarks to the Author)

Overall comment

Ebrahimi and colleagues present a comprehensive GWAS of head and neck cancer across multiple population ancestries and tumour subsites. Through analysis of genotyping data from 19,073 cases and 38,857 controls, 18 novel risk loci are identified and 6 previously identified loci are validated. In general this is an impressive study that provides by far the largest analysis of genetic susceptibility to HNSCC to date, particularly including non-european ancestries and adds to the catalogue of risk loci for this disease. There are some issues with the association study design and data presentation detailed specifically below.

Major comments

- Abstract

o "...identified 29 independent novel loci" – this appears to include the 18 genome-wide significant loci identified in the initial GWAS as well as 11 from the further HLA-finemapping analysis which utilised a less stringent significance threshold? It would be more correct to separately detail numbers that were genome-wide significant versus those that were "HLA-wide" significant

- Power calculations

o What is the estimated power of the study to detect associations in total case/control series as well as by ancestry and subsite?

- Meta-analysis/association-testing strategy

o The reported association testing strategy in general is unclear and would benefit greatly from improved clarity of presentation and description. It appears currently unclear the exact association test carried out in METAL for the "meta-analysis", "european" and "mixed" reported results, and whether this is just an association test carried out by pooling all samples together irrespective of study origin.

o While it is appreciated that testing is being carried out across different population groups, tumour sites and other clinical characteristics to discover potentially novel signals, in general it is often unclear how specific associations are to the respective groups, or whether some associations just missed the genome-wide threshold

o Accepting that the genome-wide significance threshold of 5×10^{-8} is perhaps overly conservative, this study is effectively carrying out many GWAS not just one, while reporting those significant after the same threshold. Additionally, additional fine-mapping of the HLA region further increases the number of tests undertaken

- Figure 1

o More helpful to plot both known and novel risk loci (e.g. in different colours) rather than just novel signals

o What are the Manhattan plots being displayed – are they the meta-analysis P-values or specific ancestries?

- Heritability estimation

o How do heritability estimates vary between populations? Were there substantial differences in contribution of HLA variants to heritability of HPV(+) OPC relative to other tumour types?

- Functional annotation of detected variants

o Prediction of likely "causal" variants e.g. – the authors should explore methods such as SuSiEx (PMID: 39187616), XMAP (PMID: 37898663) or PAINTOR (PMID: 38213821)

o In addition to eQTLs, it would be valuable to consider splice-QTLs and methylation-QTLs using publicly available databases

o The authors should consider utilising resources such as ENCODE, epigenome roadmap to consider overlap of association signals with tissue-specific enhancers, for example to assess consistency with subsite-specific associations

- rs78378222 risk variant at 17p13.1 (TP53)

o Figure 2B – rs78378222 T>G associated with decreased TP53 expression in whole blood. How does this compare in other tissue types, and is this consistent with the proposed tissue-specific mechanism of action that explains the protective effect in breast cancer and HNSCC but increased risk in many other cancer types (Discussion p12 lines 375-390). While an intriguing hypothesis, this is based on experiments in mice and there is a current lack of supporting human functional data, which would be valuable to include if available

o Figure 2C – How do the predicted overlapping miRNAs compare with miR-325 and miR-382 detailed in the discussion?

o Have the authors examined databases such as IsomiR-eQTL (https://data.eresearchqut.net/IsomiR_eQTL/index.html) to search for potential association between rs78378222 genotype and miRNA expression?

Minor comments:

- The full cytoband position should be provided for risk loci (e.g. 5p15.33 rather than 5p15)

- Some typographical errors e.g. p8 line 261

- Heritability analysis – p26 lines 846-848: "Univariate GREML was used to estimate heritability and was transformed onto the liability scale using global prevalence estimates from GLOBOCAN" – what prevalence estimates were used?

- Colocalisation analysis - p27 lines 859-861: "In the analyses, we considered eQTLs coinciding with genomic loci identified in this study at $p < 3.9 \times 10^{-10}$ in whole blood data; $p < 2.5 \times 10^{-7}$ in tissue data to be considered as significant)" – why were these thresholds chosen?

- Discussion p13 line 413-414 – what is meant by "smoking heaviness"?

Reviewer #3

(Remarks to the Author)

Comments for Author

In this manuscript entitled "Cross-ancestral GWAS identifies 29 novel variants across Head and Neck Cancer subsites", Ebrahimi and colleagues studied susceptibility loci for developing HNSCC. Previous GWAS studies were primarily European focused; however, Ebrahimi included data from Europe, North America, South America, South Asia, and the Middle East. While there is consensus on the major risk factors for HNSCC and there are some loci identified making a person more susceptible for developing cancer, we still cannot fully identify those that develop HNSCC. Ebrahimi found novel loci and confirmed existing loci which are increasing or even decreasing the risk for developing HNSCC. They performed robust analyses, and importantly, separated their analysis per HNSCC anatomical site. Their work is highly relevant for the field.

- Multiple susceptible loci are associated with certain risk factors, as for instance 4q23 locus (ADH1B, ADH7) is linked to genes involved in alcohol metabolism. Did the authors find any association with other risk factors next to smoking and alcohol? For example, air pollution, ageing, and poor oral hygiene. Also, are certain loci associated with sex?

- The authors identified among others the novel variant rs6679311 near MDM4, a strong negative regulator of p53, associated with increased risk. As in these individuals p53 function is already impaired, would this mean that TP53 somatic mutations will be less frequent or necessary? Is the rs6679311 variant associated with HNSCC cases with wild-type TP53? How germline susceptibilities are related to somatic mutations should at least be discussed. (Liu et al. Cancer Research Germline cancer gene expression quantitative trait loci are associated with local and global tumor mutations).

- Within HPV-negative HNSCC there is a molecular subclass, with only few or missing copy number alterations (CNAs). This group was initially found in a small cohort and later recognized in The Cancer Genome Atlas (Cancer Genome Atlas Network Nature 2015 Comprehensive genomic characterization of head and neck squamous cell carcinomas). This group is recently in more depth characterized using an OC cohort (Muijlwijk et al. Nature Communications 2024 Hallmarks of a genomically distinct subclass of head and neck cancer). Are certain loci enriched in this group? Or are certain loci associated with having less CNAs? This CNA-quiet group should at least be discussed as it is an important subclass within HPV-negative HNSCC. The authors make the important distinction between anatomical sites as well as between HPV-negative and HPV-positive OPC. In addition, the distinction between CNA-quiet and CNA-other is important to discuss. When feasible, susceptible loci should be compared between those groups.

- In the discussion the authors mention that the three novel amino acid changes which are identified (DRB1 37Asn/Ser, DRB1 233Thr, and HLA-B 67Cys/Ser/Tyr) allow for better detection of risk for HPV+ OPC across populations and might be easier to incorporate into screening strategies at the population level. It is not clear to what the authors are comparing this screening to.

- Figure 1a: overall HNSCC is overall HPV- HNSCC supposedly? Without HPV+ OPC? Also counts for Fig 2b. Not entirely clear.

- In the sentence in the introduction "Previous GWAS were limited in sample size for HNSCC subsites making interference between subsites, particularly for HLA, difficult." Please add the references again. The same goes for the sentence following. Do these limitations account for all five previously mentioned GWAS studies, if not, refer to those specific ones. And the same for the first sentence of paragraph "Refining previously identified HNSCC risk variants" in the results. The authors mention previous GWAS of HNSCCs, references either specific per loci, or all at the end, are missing.

- References are also missing in the statement "This is especially important considering the oral microbiome as a potential emerging risk factor for oral cavity cancer." In the discussion.

- In the second paragraph of the results, abbreviations are used while they were not introduced yet. EUR, AFR, and AMR in this sentence: "The T>G allele frequency of rs78378222 is 0.01 in EUR, 0.002 in AFR and AMR populations." Also, in the discussion the abbreviation eQTL was used without introducing.

- Some typos:

o In the results: "... , however this is the first time this variant was identified in within specific subsites."

o In the discussion: "GWA"

- The first paragraph within "Distinct interactions of smoking and alcohol use with risk variants" is not clear. I suppose for the first sentence the reference to figure 3a is missing, which makes it confusing.

Version 1:

Reviewer comments:

Reviewer #1

(Remarks to the Author)

Most of my concerns have been addressed in the revised version of the manuscript by Dr. Dudding et al. However, I have a few additional comments and suggestions for the authors.

1. What is the rationale for the number of principal components (PCs) selected for adjustment in the GWAS analysis? Further clarification on how the number was determined would enhance the transparency and reproducibility of the analysis.
2. The current results presented in Figure 4 and the supplementary figures illustrate SNP effects across different subgroups,

reflecting stratified analysis outcomes.

It may be informative to further evaluate whether these effects exhibit synaptic or interaction effects across subgroups, for example, SNP×smoking interaction.

3. The colocalization analysis should include all SNPs, rather than limiting the analysis to those with eQTL FDR < 0.05.

4. Colocalization analysis between mQTL and GWAS data could also be considered.

Reviewer #2

(Remarks to the Author)

Thank you to the authors for their considered responses and addressing the comments raised in the initial review. The majority of comments raised have been satisfactorily addressed, however a small number of minor points remain:

- Please provide full cytoband position for loci e.g. line 144 "4q23", line 145 "5p15"

- mQTL analysis (ST9) – it would be helpful to indicate direction of effect between GWAS SNP and methylation QTL

- Methods – helpful to de-abbreviate "PM-plots" (line 934)

- Thank you for providing power calculations – it would be valuable to include these in the manuscript as supplementary material

Reviewer #3

(Remarks to the Author)

In the manuscript "Cross-ancestral GWAS identifies 29 novel variants across Head and Neck Cancer subsites," Ebrahimi et al. investigated genetic susceptibility to HNSCC using diverse cohorts from Europe, North America, South America, South Asia, and the Middle East. Building on previous European-focused GWAS, they identified new and confirmed known risk and protective loci. Their robust, site-specific analyses provide valuable insights into HNSCC risk and advance the field significantly.

The authors have adequately addressed all prior concerns. I do not have any further remarks.

Version 2:

Reviewer comments:

Reviewer #1

(Remarks to the Author)

All of my previous questions have been fully addressed. I appreciate the authors' thorough responses and revisions. I have no further comments.

Reviewer #2

(Remarks to the Author)

Thank you to the authors for addressing my previous comments in the revision, I have no further comments and am happy to recommend the manuscript for publication.

REVIEWER COMMENTS

Response to reviewer comments for “Cross-ancestral GWAS identifies 29 novel variants across Head and Neck Cancer subsites”

Reviewer #1 (Remarks to the Author): Expert in head and neck cancer genetics and HLA genes

The authors performed a multi-ancestry GWAS and refined the genetic map of head and neck cancer as well as its subsites. Novel variants were found, additionally advancing the understanding of genetic factors on HNSCC development. I have some concerns for the study design and methods for this study.

1. Study Design and Population:

(1) This study includes samples genotyped using 9 different arrays across 19 studies, with some studies containing only case or control samples (e.g., the Sudbury study and the HN5000 study). This creates a challenge in constructing a batch effect variable that accounts for both array and study effects while preserving the distinction between cases and controls. The minor allele frequencies of the identified SNPs in cases and controls from different studies should be further examined to ensure that the observed associations are not influenced by batch effects.

We appreciate the reviewer’s important concern regarding potential batch effects. Prior to analysis, we conducted several quality control steps and sensitivity analysis to ensure that observed associations are not influenced by batch effects. While the Sudbury study did not have controls, there were controls included from Toronto with similar population demographics included. As there were no other UK-only based controls to compare with the HN5000 study, we used the ALSPAC study. Double imputation was performed (details in methods, lines 785-800) for these two to account for batch effects. We next generated a custom variable for inclusion in the logistic regression models specifically focused on addressing batch effects. This “Batch” variable -generated from both study and array data- was included to ensure that batch-related variation is properly modelled in our analysis. We further ran heterogeneity tests across batches and excluded SNPs with significant heterogeneity which resulted in the removal of 137 markers in the European sample GWAS (Methods lines 853-858). This Batch variable was then incorporated as a covariate in all association analysis to adjust for batch effects.

In response to Reviewer 1’s request, we additionally examined MAF distributions for the risk loci we identified in our study across the different imputation batches in cases and controls separately. The mean and standard deviation of the top SNPs across all imputation batches are shown here, with the small SDs highlighting the consistency of MAFs across batches:

Table. Mean and Standard Deviation Across all Imputation Batches

SNP	Cases mean (SD)	Controls mean (SD)
chr1:204493484:G:A	0.1746 (0.0099)	0.1898 (0.0108)
chr1:204590548:T:C	0.7031 (0.0103)	0.7262 (0.0142)
chr2:20677150:T:TA	0.0084 (0.0016)	0.008 (0.0019)

chr3:46468719:C:T	0.3842 (0.0263)	0.3765 (0.0218)
chr4:99318162:T:C	0.9646 (0.0177)	0.9507 (0.0213)
chr4:99335007:TC:T	0.2239 (0.0134)	0.2393 (0.0172)
chr4:99337969:T:C	0.2239 (0.0133)	0.2393 (0.0171)
chr4:99423706:C:T	0.3351 (0.0149)	0.3258 (0.0203)
chr4:99430279:C:A	0.083 (0.003)	0.1031 (0.0116)
chr4:99432854:A:T	0.1867 (0.0082)	0.2062 (0.0132)
chr4:99462539:A:G	0.0838 (0.0036)	0.1039 (0.0122)
chr5:1282204:C:A	0.3385 (0.0128)	0.3252 (0.0143)
chr5:1309789:G:A	0.5949 (0.0178)	0.5986 (0.0083)
chr5:1325475:T:C	0.4331 (0.009)	0.4282 (0.0163)
chr5:1361566:G:A	0.3243 (0.0073)	0.3324 (0.0147)
chr5:139465014:T:C	0.8491 (0.0156)	0.8634 (0.0119)
chr5:139478334:T:C	0.8459 (0.0138)	0.8606 (0.0112)
chr6:29839124:C:G	0.617 (0.0271)	0.6248 (0.0244)
chr6:29845603:C:T	0.2518 (0.0208)	0.2396 (0.0181)
chr6:30152113:G:T	0.0744 (0.0162)	0.0826 (0.0116)
chr6:30184177:C:T	0.0463 (0.0089)	0.0542 (0.0059)
chr6:31070111:TTTTTC:T	0.0484 (0.0123)	0.0384 (0.0091)
chr6:31410016:T:A	0.2262 (0.0372)	0.251 (0.0349)
chr6:31410245:T:C	0.2025 (0.0262)	0.2242 (0.015)
chr6:31462582:A:G	0.0778 (0.0368)	0.0836 (0.0273)
chr6:31566203:G:A	0.0406 (0.0141)	0.0314 (0.0108)
chr6:32613805:C:G	0.5694 (0.0245)	0.5497 (0.0242)
chr6:32619336:G:A	0.7698 (0.0186)	0.7416 (0.0157)
chr6:32619380:A:G	0.1258 (0.0167)	0.1401 (0.0184)
chr6:32620608:T:A	0.3847 (0.0473)	0.3626 (0.0293)
chr6:32634705:T:C	0.7741 (0.0177)	0.7502 (0.0171)
chr6:32700277:T:TA	0.8038 (0.0119)	0.7768 (0.0186)
chr6:33149378:A:G	0.1201 (0.0159)	0.1316 (0.0078)
chr6:148061934:G:A	0.0193 (0.0046)	0.0172 (0.0048)
chr9:21974219:A:G	0.3632 (0.0248)	0.3511 (0.0262)
chr11:81037815:C:T	0.0123 (0.0019)	0.0109 (0.0025)
chr11:132728232:G:A	0.0081 (0.0026)	0.0075 (0.0027)
chr11:133936692:G:A	0.0282 (0.0028)	0.026 (0.0038)
chr12:47926100:A:G	0.0219 (0.0064)	0.0197 (0.0052)
chr12:111648699:A:G	0.216 (0.0337)	0.1994 (0.0253)
chr12:111735644:A:G	0.1975 (0.0197)	0.1831 (0.015)
chr12:111935285:T:C	0.1944 (0.0099)	0.1791 (0.0048)
chr13:32394413:G:A	0.0139 (0.0015)	0.0073 (0.0018)
chr13:32398489:A:T	0.0139 (0.0016)	0.0073 (0.0018)
chr13:32399139:A:G	0.25 (0.0123)	0.2686 (0.0148)
chr14:25417834:CT:C	0.0152 (0.0024)	0.014 (0.0035)
chr15:49493719:A:G	0.4002 (0.0226)	0.39 (0.0285)
chr17:7655719:C:T	0.2084 (0.0072)	0.1965 (0.0095)

chr17:7668434:T:G	0.0065 (0.002)	0.0106 (0.0024)
chr17:40945241:G:T	0.0233 (0.0122)	0.0183 (0.0106)
chr19:17280519:G:A	0.3056 (0.0171)	0.2825 (0.013)
chr19:17291286:GT:G	0.3127 (0.0101)	0.2886 (0.005)
chr20:63704213:GA:G	0.9141 (0.0172)	0.9097 (0.0185)
chr20:63709883:T:C	0.9154 (0.0113)	0.9128 (0.0101)
chr22:21778123:A:C	0.008 (0.0022)	0.0061 (0.0009)

This data is also visualized below in Reviewer_Figure1, where blue bars show MAFs for cases for each Batch and red bars show MAFs for controls.

The information regarding consistency of SNPs across batch within cases and controls has now been added to the methods (lines 859-860) and this table has been added to the Supplementary Material as Supplementary Table 16.

(2) The authors used UK biobank samples but included only 14435 controls. How were these controls selected?

The controls from UKBiobank were selected carefully to balance computational efficiency with statistical power. While the ‘rule of thumb’ is that power plateaus after a 4:1 ratio of controls to cases, there is also evidence that increases in power with ratios of 10:1 can be achieved when many associations are tested, as in the case of GWAS¹. Therefore, after excluding UKBiobank participants with a previous cancer diagnosis or missing data on key variables such as smoking or alcohol use, 10 controls were selected randomly for each HNSCC case in UKBiobank. These details have been added into the methods section on lines 763-767.

2. Analysis:

(1) The author performed cross-ancestry meta-analysis by using a fixed effect model in METAL. I think a random effect model like MR-MEGA should be used to consider the different standard errors of SNPs in different ancestries.

We agree with the reviewer that methods that use random effects such as MR-MEGA are useful to evaluate effects of SNPs more powerfully across different ancestries. However, we

recognize the limitation of our study: that while we have included multiple ancestries, they individually do not have enough power to evaluate SNPs that may have heterogeneous effects across ancestries. Therefore, our focus is on identifying variants with homogeneous effects across populations and as such, we used a fixed effects model to support this assumption. In addition, due to this limitation, we separated participants of European ancestry with those of all others (“Mixed”). This approach maximized statistical power by combining small numbers across ancestral groups, while the principal components simultaneously adjusted for population structure within this group, allowing for robust estimates as evidenced by the Q-Q plots in Supplementary Figure 15. Our approach of meta-analyzing only 2 groups, further supported the use of the fixed effect. Random-effect models require a sufficient number of groups to accurately estimate group-level variation (ex: MR-MEGA requires at least 3 groups) to avoid biased or imprecise estimates. Finally, our Cochran’s Q-test for heterogeneity indicated minimal variability across ancestries (Supplementary Figure 7), reducing any additional benefit that random-effects models may have over fixed effects model².

(2) Statistical Analysis: Pleiotropic effect and site-specific effect of SNPs could be explored, for example, by ASSET and metasoftware, and be better presented.

We thank the reviewer for this comment. Our approach for evaluating pleiotropy and site-specific effects was based on identifying the same or linked SNPs (defined by LD) across subsites. However, we did not consider a meta-analysis across subsites as done by the tools highlighted by the reviewer and greatly appreciated this suggestion. We’ve implemented metasoftware which we feel adds a different perspective to our results and further helps to explain individual site contributions to risk loci identified from the overall HNC analysis. In addition, our pleiotropic and site-specific results are further supported by this new method and we have integrated its results into our results section in lines 269-279, into Figures 2 and 3, and into Supplementary Figures 8.

(3) The results of the previously identified HNSCC risk variants in this study should be presented.

The results of the previously identified HNSCC risk variants, the references that originally identified them, their effect size in our study and resulting p-values can all be found in Supplementary Table 3. The decision to keep only the novel variants in the main table was driven by space considerations of the journal. However, we recognize that this is important to present in the main text and incorporated this information where relevant throughout the text (example: lines 176-177). In addition, we have a specific section in the results entitled “Refining previously identified HNSCC risk variants” which discusses how our study contributes to further understanding these previously identified risk variants.

(4) In addition to eQTL data, other functional annotation should be performed. For example, whether the identified SNPs are located in the chromatin/transcription active region, how they regulate the target SNPs.

We thank the reviewer for this suggestion. We agree that additional functional annotation would help to characterize our variants. We’ve now implemented a comprehensive annotation of the identified SNPs using multiple resources to assess their regulatory potential. We first utilized ENCODE/SCREEN (<https://screen.wenglab.org/>), which

integrates ATAC-seq, DNase-seq, and histone modification (e.g. H3K27ac, H3K4me3) data to determine whether the SNPs reside in open chromatin regions and active regulatory elements. Next, we applied RegulomeDB (<https://regulomedb.org/regulome-search>) to score the SNPs based on multiple lines of evidence (e.g. DNase hypersensitivity, transcription factor binding, and histone marks). Variants with top ranks indicate a strong likelihood of regulatory function. Finally, we used ForgeDB (<https://foragedb.cancer.gov/>) to assess SNP enrichment in tissue-specific regulatory elements and to predict enhancer–promoter interactions. We applied this to all top risk loci and also any high-LD proxy variants. We did also run HaploReg but as this output overlapped with annotations from above, we provide annotations to variants using ENCODE/SCREEN, RegulomeDB and ForgeDB. These annotations are in the new Supplementary Table 5 and referred to in the results.

(5) The interaction between SNPs and smoking/drinking should be further investigated. Specifically, it would be valuable to explore whether the effects of smoking and drinking differ among individuals with varying genetic risk profiles. Such an analysis could offer valuable insights into personalized prevention strategies for HNSCC.

We agree with the reviewer that the interaction between SNPs and smoking/drinking are interesting for further investigation. To evaluate the effect of our SNPs with regard to these exposures, we conducted a stratified analysis to evaluate the effects of SNPs among: 1) ever smokers and ever drinkers; 2) ever smokers and never drinkers; 3) never smokers and ever drinkers; and 4) never smokers and never drinkers. We identified 3 main SNPs of interest from this analysis. 2 independent SNPs in BRCA2 and 1 in ADH1B. The results section entitled, “Distinct interactions of smoking and alcohol use with risk variants” describes these results.

To address this reviewer’s comments, we have additionally added a Supplementary Figure 9 to highlight the effects of smoking/drinking across these 4 groups across the genetic risk SNP profiles and referenced these in the results section line 282. We have also highlighted in our discussion section that future studies should evaluate this aspect of personalized prevention more comprehensively, such as in the context of polygenic risk scores on lines 439-440.

(6) Given the complexity of HLA region, conditional analysis on the previously reported HLA variants should be performed to confirm the “novel” role of the newly identified HLA variants.

We thank the reviewer for this comment as this is an important point to clarify to our readers. Our novel HLA variants were defined by 1) independence from lead variants reported across subsites; and 2) Independence from previously reported variants. There were several thresholds that defined independence. First, we ran conditional analysis and if the variant tested remained associated with risk, it was deemed independent. However, if the p-value of the test variant after conditioning dropped close to the threshold but didn’t pass it, we also checked LD. If LD indicated low or no correlation ($R^2 < 0.3$), the variants were deemed independent. If LD was high and the conditional p-value was marginal or above the threshold, the variants were deemed dependent. These criteria are defined in the Methods lines 921-928 and shown in Supplementary Table 10.

However, in responding to this comment, we realize that this was not overtly stated in the main text outside the methods and that table S10 was slightly confusing. Therefore, we have clarified our definition of “novel” for the HLA variants in the main text results section on

lines 318-319 with a reference to table S10. We also revised Supplementary table 10 for clarity, readability and interpretation.

(7) Are the allele frequencies and the LD structure of the identified HLA variants, particularly those ancestry specific, different among different ancestral groups?

In the main text, we focused on the major differences of the AF of the haplotype associated with HPV-positive OPC and of the novel amino acid changes we identified, that comprise the haplotype across ancestries (Figure 5d). For the other variants, difference in AF across ancestral groups really depends on the variant. We thank the reviewer for this question as we agree it would be interesting to report this. Therefore, we have amended table S12 which reported all the HLA variants to include AFs across each of the ancestries included in the analysis for comparison.

(8) How much is the contribution of rare and common variants, HLA and non-HLA variants, the reported variants and the novel variants, to the heritability of HNSCC and different subtypes?

We appreciate this suggestion. While our study was focused primarily on common variants, we have now evaluated the contribution of HLA variants to HNSCC heritability and its subsites. We quantified the proportion of variance in cancer risk explained by the sentinel variants in the HLA region identified from the GWAS, thereby evaluating their relative contribution to the total variance explained. We've added this information into lines 947-956 of the methods and lines 277-279 in the results.

(9) The EAF of some of the 18 novel variants are not shown in Table 1. Please provide the detailed information.

Thank you for pointing out this oversight. We have now provided this information in both Table 1 and Supplementary Table 3.

(10) Experimental validation should be performed to confirm the effect of SNP on gene expression, for example, rs78378222 on TP53 expression and its effect on the impairment of miRNA binding, rs28419191 and rs1131769 on CTNNA1 expression.

We agree that it would be interesting to perform experimental validation on the biological function of our top SNPs. However, unfortunately, we do not have RNA on these samples nor any expression data. To mitigate this limitation, we have added a comprehensive functional annotation in addition to the eQTL analysis to further contextualize the biological relevance of the identified SNPs. In addition, we interrogated expression data from the eQTLGen Consortium and Genotype-Tissue Expression (GTEx, version 8) to evaluate associations of SNPs on gene expression. These findings are reported in our results on lines 197-200.

3. Explanation and others:

(1) How do the authors explain that the heritability of HPC+ oropharyngeal cancer is significantly higher than that of HPC- oropharyngeal cancer? It appears that virological factors may contribute more to the development of HPC+ oropharyngeal cancer.

We also feel the difference in heritability of HPV+ OPC and HPV- OPC is interesting and have investigated this further. In doing so we identified an error in the prevalence estimates used in the GREML analysis and when corrected the substantial difference in heritability are reduced. Further to this, GREML heritability estimates can be biased when strongly associated variants are found in an area of the genome with different LD properties to the rest of the genome, as is the case with the HLA region in HPV+ OPC³. We have therefore used an alternative method to estimate heritability which is less affected by this bias (Linkage disequilibrium score regression⁴). These results are updated in lines 178-180, and methods lines 947-955

(2) Biological explanation should be included for the site specific or HPV+/HPV-specific SNPs.

We agree with the reviewer that biological explanation for identified SNPs is important. We have tried to suggest biological explanations for all variants identified within the limits of the scope of the paper and word count. In addition, we include a supplementary table with biological descriptions of all SNPs (Supplementary Table 4) and extensive functional annotation (Supplementary Table 5).

(3) The results section of the manuscript should be organized in a clearer and more concise manner.

Given the extensive reviewer comments, we have reviewed and revised the results section for clarity and brevity.

Reviewer #2 (Remarks to the Author): Expert in cancer genetics and GWAS

Overall comment

Ebrahimi and colleagues present a comprehensive GWAS of head and neck cancer across multiple population ancestries and tumour subsites. Through analysis of genotyping data from 19,073 cases and 38,857 controls, 18 novel risk loci are identified and 6 previously identified loci are validated. In general this is an impressive study that provides by far the largest analysis of genetic susceptibility to HNSCC to date, particularly including non-european ancestries and adds to the catalogue of risk loci for this disease. There are some issues with the association study design and data presentation detailed specifically below.

Major comments

- Abstract

o “...identified 29 independent novel loci” – this appears to include the 18 genome-wide significant loci identified in the initial GWAS as well as 11 from the further HLA-finemapping analysis which utilised a less stringent significance threshold? It would be more correct to separately detail numbers that were genome-wide significant versus those that were “HLA-wide” significant

We agree with the reviewer that this would be more representative of the study design and thresholds and have amended this in the abstract in lines 111-112 and in the discussion.

- Power calculations

o What is the estimated power of the study to detect associations in total case/control series as well as by ancestry and subsite?

Power calculations to detect Odds Ratios based on MAF across ancestry and subsite are presented below:

Power Calculations						
Ancestry	Cancer site	MAF	power	Sample Size	Case Proportion	Detectable OR
European	All cases	0.005	0.8	46468	0.3	1.8
		0.01	0.8	46468	0.3	1.5
		0.05	0.8	46468	0.3	1.2
		0.1	0.8	46468	0.3	1.2
		0.25	0.8	46468	0.3	1.1
	Oral Cavity	0.005	0.8	35344	0.1	2.5
		0.01	0.8	35344	0.1	2.0
		0.05	0.8	35344	0.1	1.4
		0.1	0.8	35344	0.1	1.3
		0.25	0.8	35344	0.1	1.2
	Larynx	0.005	0.8	34620	0.08	2.6
		0.01	0.8	34620	0.08	2.1
		0.05	0.8	34620	0.08	1.4
		0.1	0.8	34620	0.08	1.3
		0.25	0.8	34620	0.08	1.2
	Hypopharynx	0.005	0.8	32399	0.02	4.7
		0.01	0.8	32399	0.02	3.4
		0.05	0.8	32399	0.02	2.0
		0.1	0.8	32399	0.02	1.7
		0.25	0.8	32399	0.02	1.5
	HPV(+) Oropharynx	0.005	0.8	20146	0.1	3.1
		0.01	0.8	20146	0.1	2.4
		0.05	0.8	20146	0.1	1.5
		0.1	0.8	20146	0.1	1.4
		0.25	0.8	20146	0.1	1.3
HPV(-) Oropharynx	0.005	0.8	19114	0.05	4.1	
	0.01	0.8	19114	0.05	3.0	
	0.05	0.8	19114	0.05	1.8	
	0.1	0.8	19114	0.05	1.6	
	0.25	0.8	19114	0.05	1.4	
Mixed	All cases	0.005	0.8	11462	0.5	3.7
		0.01	0.8	11462	0.5	2.4
		0.05	0.8	11462	0.5	1.5
		0.1	0.8	11462	0.5	1.3

		0.25	0.8	11462	0.5	1.2
	Oral Cavity	0.005	0.8	7974	0.3	4.2
		0.01	0.8	7974	0.3	2.8
		0.05	0.8	7974	0.3	1.6
		0.1	0.8	7974	0.3	1.4
		0.25	0.8	7974	0.3	1.3
	Larynx	0.005	0.8	7511	0.2	4.5
		0.01	0.8	7511	0.2	3.0
		0.05	0.8	7511	0.2	1.7
		0.1	0.8	7511	0.2	1.5
		0.25	0.8	7511	0.2	1.3
	Hypopharynx	0.005	0.8	6221	0.05	8.3
		0.01	0.8	6221	0.05	5.3
		0.05	0.8	6221	0.05	2.6
		0.1	0.8	6221	0.05	2.1
		0.25	0.8	6221	0.05	1.7
Additive models with p-value threshold of 5e-08						

- Meta-analysis/association-testing strategy

o The reported association testing strategy in general is unclear and would benefit greatly from improved clarity of presentation and description. It appears currently unclear the exact association test carried out in METAL for the “meta-analysis”, “european” and “mixed” reported results, and whether this is just an association test carried out by pooling all samples together irrespective of study origin.

As we have individual-level data, we conducted a mega analysis within the ancestral groups and then meta-analyzed across the 2 ancestry groups. We ensured that effects due to study origin, study design and array used were accounted for within ancestry with our focused efforts on developing the Batch variable. We’ve clarified this association testing strategy in the methods, lines 847-848. We’ve also added a reference to Supplementary Figure 12 here where this is described in a schematic.

o While it is appreciated that testing is being carried out across different population groups, tumour sites and other clinical characteristics to discover potentially novel signals, in general it is often unclear how specific associations are to the respective groups, or whether some associations just missed the genome-wide threshold

Table 1, Table 2, Supplementary Table 3 and Supplementary Table 10 describe the associations specific to their respective groups. We’ve also organized the results by cancer subsite. However, we recognize that in the results where we tried to contextualize these top hits with reference to others in LD or across subsites. this may become unclear. We have gone through the results to clarify our wording. For example, we’ve specified the cross ancestral novel variants from the ancestry-specific variants throughout the results section.

o Accepting that the genome-wide significance threshold of 5×10^{-8} is perhaps overly conservative, this study is effectively carrying out many GWAS not just one, while

reporting those significant after the same threshold. Additionally, additional fine-mapping of the HLA region further increases the number of tests undertaken

We thank the reviewer for this thoughtful comment. While we acknowledge that the conventional genome-wide significance threshold of 5×10^{-8} may be considered conservative, we respectfully believe that it remains appropriate for our analysis. Our study involves multiple GWAS; however, these analyses are not entirely independent. For example, the GWAS share a common set of controls across the different subsites, and many of the variants demonstrate overlapping risk profiles. A case in point is the stop-gained variant in BRCA2, which leads to a premature stop codon 93 amino acids into the protein and is associated with risk for both hypopharynx and larynx cancers. This overlap in genetic risk factors implies that the tests are correlated to a degree that reduces the need for a more stringent multiple testing correction.

We have carefully considered the suggestion to adjust the significance threshold further, but given these interdependencies, we believe that maintaining the threshold at 5×10^{-8} is justified for the purposes of this study. We have revised the manuscript in the discussion in lines 486-488 to more clearly articulate this caution.

- Figure 1

o More helpful to plot both known and novel risk loci (e.g. in different colours) rather than just novel signals

Figure 1 has been updated to include both novel and known risk loci in different colors.

o What are the Manhattan plots being displayed – are they the meta-analysis P-values or specific ancestries?

The main figure 1a only contains the meta-analysis plots as described in the figure caption. Initially we included the positions of loci identified from specific ancestries but from this reviewer's questions, we realized this was confusing and now only show the novel and known loci from the meta-analysis.

- Heritability estimation

o How do heritability estimates vary between populations? Were there substantial differences in contribution of HLA variants to heritability of HPV(+) OPC relative to other tumour types?

We did estimate heritability across both ancestry samples but chose not to present the heritability estimates from the mixed ancestry sample in the main text as we felt these were likely inaccurate due to low sample size and the heterogeneous nature of this sample. We do include these in Supplementary Table 6 however. We have conducted a comparison of the variance explained by HLA variants across cancer subsites. We've added this information into lines 948-956 of the methods and lines 276-279 in the results.

- Functional annotation of detected variants

o Prediction of likely “causal” variants e.g. – the authors should explore methods such as SuSiEx (PMID: 39187616), XMAP (PMID: 37898663) or PAINTOR (PMID: 38213821)

We agree with the reviewer that functional annotation of detected variants should be improved. While we believe that an extended analysis on causal variants would be beyond the scope of the paper, we have added a comprehensive annotation and refer to our changes documented in the response to Reviewer 1's comment number 4.

o In addition to eQTLs, it would be valuable to consider splice-QTLs and methylation-QTLs using publicly available databases

We have now added results on splice-QTLs and methylation-QTLs in whole blood, esophagus, and lung tissue and incorporated them into our colocalization analysis. The updated results have been added to Supplementary Table S8. In addition, methylation QTLs (mQTLs) overlapping the identified genomic loci at genome-wide significance ($P < 5 \times 10^{-8}$) are detailed in Supplementary Table S9. Notably, we identified new sQTL evidence for TP53 and LIME1, along with moderate associations at several CpG sites linked to LIME1 (cg01176363, cg05643964, cg02966332) and SLC2A4RG (cg07196408). These results have been added into the results, for example on lines 218-220.

o The authors should consider utilising resources such as ENCODE, epigenome roadmap to consider overlap of association signals with tissue-specific enhancers, for example to assess consistency with subsite-specific associations

We agree with the reviewer that the functional annotation of variants needed to be further explored. We have added a comprehensive annotation and refer to our changes documented in the response to Reviewer 1's comment number 4.

- rs78378222 risk variant at 17p13.1 (TP53)

o Figure 2B – rs78378222 T>G associated with decreased TP53 expression in whole blood. How does this compare in other tissue types, and is this consistent with the proposed tissue-specific mechanism of action that explains the protective effect in breast cancer and HNSCC but increased risk in many other cancer types (Discussion p12 lies 375-390). While an intriguing hypothesis, this is based on experiments in mice and there is a current lack of supporting human functional data, which would be valuable to include if available

We thank the reviewer for noting this point. In response, we evaluated this variant with expression in relevant tissue types in humans. We tested effect in whole blood, which had the largest sample size ($n \sim 31,000$ in eQTLgen), in esophagus ($n \sim 500$) and lung ($n \sim 500$), which were the closest relevant tissue types. We found similar effects in whole blood and esophagus but not lung. However, given that these analyses are sensitive to both sample size and tissue specificity, this highlights the need to for further downstream studies in this area. Considering this and an ambiguity flagged in an editorial comment published since our submission, on the mouse paper cited in the discussion, we have amended our discussion item on this point on lines 407-414.

o Figure 2C – How do the predicted overlapping miRNAs compare with miR-325 and miR-382 detailed in the discussion?

This was included in the figure to contextualize the SNP but this did not come from any formal prediction analysis. As noted in the comment above, the discussion regarding miRNA has been amended in lines 407-414 and therefore comparison here is no longer applicable.

o Have the authors examined databases such as IsomiR-eQTL (https://data.eresearchqut.net/IsomiR_eQTL/index.html) to search for potential association between rs78378222 genotype and miRNA expression?

We did try to evaluate the relationship between rs78378222 and miRNA expression. However, there is no data available for head and neck cancers on these databases and thus unfortunately we were not able to evaluate this.

Minor comments:

- The full cytoband position should be provided for risk loci (e.g. 5p15.33 rather than 5p15)

Table 1 and Supplementary Table 3 have been updated with full cytoband positions.

- Some typographical errors e.g. p8 line 261

These errors have been fixed.

- Heritability analysis – p26 lines 846-848: “Univariate GREML was used to estimate heritability and was transformed onto the liability scale using global prevalence estimates from GLOBOCAN” – what prevalence estimates were used?

In response to reviewer 1 (point 3.1), we have re-calculated our heritability estimates using Linkage Disequilibrium Score Regression (LDSR) which does not use prevalence estimates, we have therefore removed the mention of prevalence estimates from the manuscript.

- Colocalisation analysis - p27 lines 859-861: “In the analyses, we considered eQTLs coinciding with genomic loci identified in this study at $p < 3.9 \times 10^{-10}$ in whole blood data; $p < 2.5 \times 10^{-7}$ in tissue data to be considered as significant)” – why were these thresholds chosen?

We apologize for this error. We tested for colocalization if a variant was related to gene expression with an FDR of ≤ 0.05 . This update has been incorporated into the manuscript in the methods on lines 967-969.

- Discussion p13 line 413-414 – what is meant by “smoking heaviness”?

Smoking heaviness was defined as cigarettes smoked per day in the manuscript where this trait was defined⁵, and we have clarified this in line 438.

Reviewer #3 (Remarks to the Author): Expert in head and neck cancer omics and genetics

Comments for Author

In this manuscript entitled ‘‘Cross-ancestral GWAS identifies 29 novel variants across Head and Neck Cancer subsites’’, Ebrahimi and colleagues studied susceptibility loci for

developing HNSCC. Previous GWAS studies were primarily European focused; however, Ebrahimi included data from Europe, North America, South America, South Asia, and the Middle East. While there is consensus on the major risk factors for HNSCC and there are some loci identified making a person more susceptible for developing cancer, we still cannot fully identify those that develop HNSCC. Ebrahimi found novel loci and confirmed existing loci which are increasing or even decreasing the risk for developing HNSCC. They performed robust analyses, and importantly, separated their analysis per HNSCC anatomical site. Their work is highly relevant for the field.

- Multiple susceptible loci are associated with certain risk factors, as for instance 4q23 locus (ADH1B, ADH7) is linked to genes involved in alcohol metabolism. Did the authors find any association with other risk factors next to smoking and alcohol? For example, air pollution, ageing, and poor oral hygiene. Also, are certain loci associated with sex?

There were 2 SNPs that showed heterogeneity by sex. rs17529509 identified in all sites and rs1229984 for oral cavity cancer risk. We have added sentences in the results to highlight this at lines 259-261. We agree with the reviewer that associations with risk factors other than smoking and alcohol would be very interesting, unfortunately, we do not have this data across the contributing studies. We added a sentence in the discussion to note that future studies should evaluate additional exposures in lines 441-443.

- The authors identified among others the novel variant rs6679311 near MDM4, a strong negative regulator of p53, associated with increased risk. As in these individuals p53 function is already impaired, would this mean that TP53 somatic mutations will be less frequent or necessary? Is the rs6679311 variant associated with HNSCC cases with wild-type TP53? How germline susceptibilities are related to somatic mutations should at least be discussed. (Liu et al. Cancer Research Germline cancer gene expression quantitative trait loci are associated with local and global tumor mutations).

We thank the reviewer for raising this interesting point. While we don't currently have somatic data available to test this, we have added the concept of how germline susceptibilities are related to somatic mutations in the discussion section lines 412-414.

- Within HPV-negative HNSCC there is a molecular subclass, with only few or missing copy number alterations (CNAs). This group was initially found in a small cohort and later recognized in The Cancer Genome Atlas (Cancer Genome Atlas Network Nature 2015 Comprehensive genomic characterization of head and neck squamous cell carcinomas). This group is recently in more depth characterized using an OC cohort (Muijlwijk et al. Nature Communications 2024 Hallmarks of a genomically distinct subclass of head and neck cancer). Are certain loci enriched in this group? Or are certain loci associated with having less CNAs? This CNA-quiet group should at least be discussed as it is an important subclass within HPV-negative HNSCC. The authors make the important distinguishment between anatomical sites as well as between HPV-negative and HPV-positive OPC. In addition, the distinction between CNA-quiet and CNA-other is important to discuss. When feasible, susceptible loci should be compared between those groups.

We thank the reviewer for raising this interesting point. While we believe that a specific analysis on the somatic molecular subclasses of oral cavity cancer is out of the scope of this paper.

- In the discussion the authors mention that the three novel amino acid changes which are identified (DRB1 37Asn/Ser, DRB1 233Thr, and HLA-B 67Cys/Ser/Tyr) allow for better detection of risk for HPV+ OPC across populations and might be easier to incorporate into screening strategies at the population level. It is not clear to what the authors are comparing this screening to.

Our comment regarding these amino acid changes was intended to be focused on the allele frequencies of these variants compared to the frequency of the haplotype that they underlie. The screening strategy was intended as an example of how this information can be potentially used, should there ever be a screening strategy. However, as the reviewer points out, there is no current screening strategy so perhaps this was not the best example to use. We have revised this section in the discussion to clarify our intention of describing utility of identifying these variants in lines 466-470.

- Figure 1a: overall HNSCC is overall HPV- HNSCC supposedly? Without HPV+ OPC? Also counts for Fig 2b. Not entirely clear.

We've clarified the text to indicate that overall HNSCC is "all sites combined" indicating that this includes all our cases, and not just HPV(-) HNSCC.

- In the sentence in the introduction "Previous GWAS were limited in sample size for HNSCC subsites making interference between subsites, particularly for HLA, difficult." Please add the references again. The same goes for the sentence following. Do these limitations account for all five previously mentioned GWAS studies, if not, refer to those specific ones. And the same for the first sentence of paragraph "Refining previously identified HNSCC risk variants" in the results. The authors mention previous GWAS of HNSCCs, references either specific per loci, or all at the end, are missing.

The GWAS references have been added into the sentences in the introduction and for the first sentence of the paragraph "Refining previously identified HNSCC risk variants" in the results.

- References are also missing in the statement "This is especially important considering the oral microbiome as a potential emerging risk factor for oral cavity cancer." In the discussion.

The references supporting the oral microbiome as an emerging risk factor for oral cavity cancer have been added in the discussion in line 460.

- In the second paragraph of the results, abbreviations are used while they were not introduced yet, EUR, AFR, and AMR in this sentence: "The T>G allele frequency of rs78378222 is 0.01 in EUR, 0.002 in AFR and AMR populations." Also, in the discussion the abbreviation eQTL was used without introducing.

We've revised to introduce the abbreviations EUR, AFR, AMR on lines 192-193 and eQTL and sQTL at their first use on line 198.

- Some typos:

o In the results: “.. , however this is the first time this variant was identified in within specific subsites.”

o In the discussion: “GWA”

The typos in line 268 was fixed. Line 394 in the discussion was not a typo, due to sentence structure we spelled out the word “studies” rather than including it in the acronym.

- The first paragraph within “Distinct interactions of smoking and alcohol use with risk variants” is not clear. I suppose for the first sentence the reference to figure 3a is missing, which makes it confusing.

We've added a first sentence to this paragraph to orient the reader to the relevant figure and purpose of this section in line 281-282.

1. Katki HA, Berndt SI, Machiela MJ, et al. Increase in power by obtaining 10 or more controls per case when type-1 error is small in large-scale association studies. *BMC Med Res Methodol* 2023; **23**(1): 153.
2. Gelman AH, J. Data Analysis Using Regression and Multilevel/Hierarchical Models. New York: Colombia University; 2006.
3. Barry CS, Walker VM, Cheesman R, Davey Smith G, Morris TT, Davies NM. How to estimate heritability: a guide for genetic epidemiologists. *Int J Epidemiol* 2023; **52**(2): 624-32.
4. Bulik-Sullivan BK, Loh PR, Finucane HK, et al. LD Score regression distinguishes confounding from polygenicity in genome-wide association studies. *Nat Genet* 2015; **47**(3): 291-5.

REVIEWER COMMENTS

Reviewer #1 (Remarks to the Author):

Most of my concerns have been addressed in the revised version of the manuscript by Dr. Dudding et al. However, I have a few additional comments and suggestions for the authors.

1. What is the rationale for the number of principal components (PCs) selected for adjustment in the GWAS analysis? Further clarification on how the number was determined would enhance the transparency and reproducibility of the analysis.

We selected the number of PCs for each GWAS based on their significant associations ($p < 0.05$) with case-control status after adjusting for sex and imputation batch. These informative PCs, along with sex and imputation batch, were included as model covariates in the GWAS analysis. This rationale is explained in the methods in line 842-845. However, we have also added this information in lines 873-874 where the models were explained, for clarity.

2. The current results presented in Figure 4 and the supplementary figures illustrate SNP effects across different subgroups, reflecting stratified analysis outcomes. It may be informative to further evaluate whether these effects exhibit synaptic or interaction effects across subgroups, for example, SNP×smoking interaction.

We thank the reviewer for this suggestion. Our rationale for presenting the results as a stratified analysis with a test for heterogeneity is we anticipated a biological interaction rather than a multiplicative statistical one. For example, the effects of the identified BRCA2 variant (rs11571833) acts only in the presence of a carcinogen (smoking or alcohol in this case). However, in view of the reviewer's comments, we have compared models with and without an interaction term between the SNP and smoking/alcohol variable and find no strong evidence of multiplicative statistical interaction beyond that expected after multiple correcting. Given it is difficult to identify if the lack of evidence for interaction is due to it not being present or lack of power, we do not feel the paper is improved if this is included.

We include a table of results (Reviewer Table 1) for the reviewer which tests for the presence of an interaction between the lead SNPs and the compound smoking/alcohol variable (never smoker, never drinker; never smoker, ever drinker; ever smoker, never drinker; ever smoker, ever drinker). We also provide the p-value for a chi-squared test comparing the fit of the models.

3. The colocalization analysis should include all SNPs, rather than limiting the analysis to those with eQTL FDR < 0.05.

We thank the reviewer for raising this important point and apologize for any confusion caused by our original description. We confirm that all SNPs (at assessed loci) were indeed used for colocalization analysis.

To clarify, this analysis was conducted in a two-stage approach. In the initial stage, we evaluated all associated SNPs within genomic loci implicated in head and neck squamous cell carcinoma (HNSCC) risk for significant associations with expression quantitative trait loci (eQTLs) and splicing quantitative trait loci (sQTLs). SNPs demonstrating nominal evidence of association (FDR-adjusted p-value < 0.05) proceeded to the second stage, where we conducted the formal colocalization analysis using all SNPs available at that locus.

We have revised the methods section accordingly:

“Colocalization analysis was performed at all genetic loci associated with HNSCC risk. Loci were considered eligible for assessment with colocalization if they harboured at least one variant significantly associated with expression or splicing (eQTL/sQTL; FDR-adjusted $p < 0.05$) or with DNA methylation levels (mQTL; $p < 5 \times 10^{-8}$). For each eligible locus, colocalization analysis was then performed using all SNPs available within the locus.” (line 979-987)

4. Colocalization analysis between mQTL and GWAS data could also be considered.

We have updated the table with colocalization analyses between mQTLs and HNSCC risk and adjusted the methods and results as appropriate. The results of these analyses are included alongside other colocalization findings in Supplementary Table 8.

Reviewer #2 (Remarks to the Author):

Thank you to the authors for their considered responses and addressing the comments raised in the initial review. The majority of comments raised have been satisfactorily addressed, however a small number of minor points remain:

- Please provide full cytoband position for loci e.g. line 144 “4q23”, line 145 “5p15”

We have updated the text to align with the cytoband positions in Supplementary Table 3.

- mQTL analysis (ST9) – it would be helpful to indicate direction of effect between GWAS SNP and methylation QTL

Thank you for the suggestion. As part of our response to comments from other reviewers, we conducted colocalization analyses between mQTLs and HNSCC risk, and included the results in Supplementary Table 8. In response to your comment, we have now updated Supplementary Table 8 to include the direction of significant associations.

- Methods – helpful to de-abbreviate “PM-plots” (line 934)

We’ve de-abbreviated PM-plots to Posterior Mean (PM)-plots in line 891 in the text.

- Thank you for providing power calculations – it would be valuable to include these in the manuscript as supplementary material

The power calculations have been added into the supplementary material as Supplementary Table 17 and referenced in the text at lines 778-779

Reviewer Table 1. Results of interaction tests

RSID	Nearest gene	No interaction term			Including interaction term			P-value interaction
		OR	LCI	UCI	OR	LCI	UCI	
rs61817953	PIK3C2B	0.92	0.86	0.98	0.93	0.87	0.99	0.988
rs58223772	ADH1B	0.92	0.87	0.98	0.96	0.90	1.02	0.266
rs58223772	ADH1C	0.93	0.87	0.98	0.96	0.91	1.02	0.257
rs1154462	ADH7	1.05	0.99	1.10	1.03	0.98	1.08	0.573
rs17529509	ADH7	0.87	0.80	0.94	0.86	0.79	0.93	0.671
rs31493	CLPTM1L	0.91	0.86	0.96	0.86	0.81	0.90	0.180
rs13215307	TRIM15	0.82	0.74	0.92	0.86	0.77	0.96	0.581
rs541752611	-	1.10	0.97	1.24	1.08	0.95	1.22	0.688
rs9266806	MICA	0.97	0.92	1.03	1.04	0.98	1.10	0.123
rs3731239	CDKN2A	1.12	1.06	1.17	1.15	1.09	1.21	0.338
rs77750788	IGSF9B	1.06	0.91	1.23	1.02	0.88	1.19	0.612
rs10774632	BRAP	1.13	1.06	1.20	1.04	0.97	1.10	0.024
rs12314527	TMEM116	1.13	1.06	1.20	1.03	0.97	1.10	0.020
rs11571815	BRCA2	2.00	1.59	2.53	2.28	1.80	2.88	0.489
rs7334543	BRCA2	0.92	0.87	0.97	0.93	0.88	0.99	0.935
rs55831773	ATP1B2	1.10	1.03	1.17	1.08	1.02	1.15	0.692
rs10419397	ANKLE1	1.11	1.06	1.18	1.13	1.07	1.19	0.674
rs6679311	MDM4	1.04	0.99	1.10	1.03	0.98	1.09	0.994
rs60622800	TERT	1.11	1.05	1.16	1.16	1.11	1.22	0.347
rs28419191	ECSCR	1.24	1.16	1.34	1.18	1.10	1.27	0.606
rs407238	HLA-G	1.18	1.12	1.23	1.17	1.11	1.23	0.329
rs9271300	HLA-DQA1	0.90	0.85	0.94	0.88	0.84	0.93	0.565
rs9282195	-	0.89	0.84	0.94	0.87	0.82	0.93	0.791
rs67351073	ZGPAT	0.83	0.75	0.91	0.83	0.75	0.91	0.246
rs138707495	GDF7	1.06	0.80	1.40	0.93	0.70	1.23	0.003
rs1520483	LTF	0.97	0.92	1.02	0.98	0.93	1.03	0.230
rs1154467	ADH7	0.93	0.87	0.99	0.94	0.88	1.00	0.535
rs77324539	ADH7	0.87	0.80	0.94	0.85	0.79	0.93	0.764
rs7726159	TERT	1.12	1.06	1.18	1.15	1.09	1.22	0.621
rs421284	CLPTM1L	1.11	1.06	1.17	1.18	1.12	1.24	0.102
rs2517878	-	1.22	1.15	1.29	1.23	1.16	1.30	0.389
rs9468692	TRIM10	0.82	0.75	0.91	0.83	0.75	0.92	0.779
rs6938453	MICA	0.94	0.89	1.00	0.97	0.92	1.03	0.676
rs4360170	MICB	1.05	0.95	1.15	1.14	1.03	1.25	0.363
rs2857601	LTA	1.06	0.93	1.20	0.99	0.87	1.12	0.735
rs28688825	HLA-DQA1	0.99	0.92	1.07	1.04	0.97	1.12	0.284
rs9271452	HLA-DQA1	0.99	0.94	1.04	0.95	0.91	1.00	0.501
rs62407809	COL11A2	0.99	0.92	1.06	0.93	0.87	1.00	0.391
rs150899739	SASH1	0.98	0.82	1.17	1.08	0.90	1.29	0.619
rs181777026	TENM4	1.12	0.89	1.42	1.45	1.15	1.82	0.272
rs181194133	OPCML	1.25	0.94	1.65	1.12	0.84	1.48	0.358
rs112726671	VDR	1.18	0.97	1.45	1.18	0.97	1.45	0.964
rs847900	ALDH2	1.14	1.07	1.22	1.05	0.98	1.12	0.037
rs11571833	BRCA2	2.01	1.59	2.54	2.28	1.80	2.88	0.489
rs200410709	STXBP6	0.87	0.71	1.07	0.95	0.77	1.16	0.807
rs12910284	FGF7	1.11	1.06	1.17	1.13	1.07	1.18	0.584
rs78378222	TP53	0.54	0.41	0.70	0.55	0.42	0.72	0.853
rs112891334	KRT23	1.23	0.99	1.51	1.29	1.05	1.59	0.420
rs72485424	MRPL34	1.11	1.05	1.17	1.13	1.07	1.19	0.605
rs577454702	MAPK1	2.08	1.48	2.93	2.17	1.54	3.05	0.324
rs1229984	ADH1B	0.65	0.55	0.76	0.76	0.64	0.89	0.161
rs1131769	STING1	1.24	1.16	1.34	1.20	1.12	1.29	0.643
rs9271376	HLA-DRB1	0.90	0.85	0.95	0.88	0.83	0.93	0.774
rs3104369	HLA-DQA1	0.89	0.84	0.95	0.85	0.80	0.90	0.542
rs4809325	ARFRP1	0.82	0.75	0.90	0.80	0.73	0.88	0.191